# APPROXIMATION ALGORITHMS FOR SPARSE PRINCIPAL COMPONENT ANALYSIS

## ABSTRACT

Principal component analysis (PCA) is a widely used dimension reduction technique in machine learning and multivariate statistics. To improve the interpretability of PCA, various approaches to obtain sparse principal direction loadings have been proposed, which are termed Sparse Principal Component Analysis (SPCA). In this paper, we present three provably accurate, polynomial time, approximation algorithms for the SPCA problem, without imposing any restrictive assumptions on the input covariance matrix. The first algorithm is based on randomized matrix multiplication; the second algorithm is based on a novel deterministic thresholding scheme; and the third algorithm is based on a semidefinite programming relaxation of SPCA. All algorithms come with provable guarantees and run in low-degree polynomial time. Our empirical evaluations confirm our theoretical findings.

## 1 INTRODUCTION

Principal Component Analysis (PCA) and the related Singular Value Decomposition (SVD) are fundamental data analysis and dimension reduction tools in a wide range of areas including machine learning, multivariate statistics and many others. They return a set of orthogonal vectors of decreasing importance that are often interpreted as fundamental latent factors that underlie the observed data. Even though the vectors returned by PCA and SVD have strong optimality properties, they are notoriously difficult to interpret in terms of the underlying processes generating the data (Mahoney & Drineas, 2009), since they are linear combinations of *all* available data points or *all* available features. The concept of Sparse Principal Components Analysis (SPCA) was introduced in the seminal work of (d'Aspremont et al., 2007), where sparsity constraints were enforced on the singular vectors in order to improve interpretability. A prominent example where sparsity improves interpretability is document analysis, where sparse principal components can be mapped to specific topics by inspecting the (few) keywords in their support (d'Aspremont et al., 2007; Mahoney & Drineas, 2009; Papailiopoulos et al., 2013).

Formally, given a positive semidefinite (PSD) matrix $\mathbf{A} \in \mathbb{R}^{n \times n}$, SPCA can be defined as follows:[1]

$$\mathcal{Z}^* = \max_{\mathbf{x} \in \mathbb{R}^n, \, \|\mathbf{x}\|_2 \leq 1} \mathbf{x}^\top \mathbf{A} \mathbf{x}, \qquad \text{subject to } \|\mathbf{x}\|_0 \leq k. \qquad (1)$$

In the above formulation, $\mathbf{A}$ is a covariance matrix representing, for example, all pairwise feature or object similarities for an underlying data matrix. Therefore, SPCA can be applied for either the object or feature space of the data matrix, while the parameter $k$ controls the sparsity of the resulting vector and is part of the input. Let $\mathbf{x}^*$ denote a vector that achieves the optimal value $\mathcal{Z}^*$ in the above formulation. Then intuitively, the optimization problem of eqn. (1) seeks a *sparse*, unit norm vector $\mathbf{x}^*$ that maximizes the data variance.

It is well-known that solving the above optimization problem is NP-hard (Moghaddam et al., 2006a) and that its hardness is due to the sparsity constraint. Indeed, if the sparsity constraint was removed, then the resulting optimization problem can be easily solved by computing the top left or right singular vector of $\mathbf{A}$ and its maximal value $\mathcal{Z}^*$ is equal to the top singular value of $\mathbf{A}$.

**Notation.** We use bold letters to denote matrices and vectors. For a matrix $\mathbf{A} \in \mathbb{R}^{n \times n}$, we denote its $(i, j)$-th entry by $A_{i,j}$; its $i$-th row by $\mathbf{A}_{i*}$ and its $j$-th column by $\mathbf{A}_{*j}$; its 2-norm by

---

[1] Recall that the $p$-th power of the $\ell_p$ norm of a vector $\mathbf{x} \in \mathbb{R}^n$ is defined as $\|\mathbf{x}\|_p^p = \sum_{i=1}^n |\mathbf{x}_i|^p$ for $0 < p < \infty$. For $p = 0$, $\|x\|_0$ is a semi-norm denoting the number of non-zero entries of $x$.

$\|\mathbf{A}\|_2 = \max_{\mathbf{x} \in \mathbb{R}^n, \|\mathbf{x}\|_2 = 1} \|\mathbf{A}\mathbf{x}\|_2$; and its (squared) Frobenius norm by $\|\mathbf{A}\|_F^2 = \sum_{i,j} A_{i,j}^2$. We use the notation $\mathbf{A} \succeq 0$ to denote that the matrix $\mathbf{A}$ is symmetric positive semidefinite (PSD) and $\mathrm{Tr}(\mathbf{A}) = \sum_i A_{i,i}$ to denote its trace, which is also equal to the sum of its singular values. Given a PSD matrix $\mathbf{A} \in \mathbb{R}^{n \times n}$, its Singular Value Decomposition is given by $\mathbf{A} = \mathbf{U}\mathbf{\Sigma}\mathbf{U}^T$, where $\mathbf{U}$ is the matrix of left/right singular vectors and $\mathbf{\Sigma}$ is the diagonal matrix of singular values.

## 1.1 OUR CONTRIBUTIONS

We present three algorithms for SPCA and associated quality-of-approximation results (Theorems 2.2, 3.1, and 4.1). All three algorithms are simple, intuitive, and run in $\mathcal{O}(n^{3.5})$ or less time. They return a vector that is provably sparse and, when applied to the input covariance matrix $\mathbf{A}$, provably captures a fraction of the optimal solution $\mathcal{Z}^*$. We note that in all three algorithms, the output vector has a sparsity that depends on $k$ (the target sparsity of the original SPCA problem of eqn. (1)) and $\epsilon$ (an accuracy parameter between zero and one).

The first algorithm is based on randomized, approximate matrix multiplication: it randomly (but non-uniformly) selects a subset of $\mathcal{O}(k/\epsilon^2)$ columns of $\mathbf{A}^{1/2}$ (the square root of the PSD matrix $\mathbf{A}$) and computes its top right singular vector. The output of this algorithm is precisely this singular vector, padded with zeros to become a vector in $\mathbb{R}^n$. It turns out that this simple algorithm, which, surprisingly has not been analyzed in prior work, returns an $\mathcal{O}(k/\epsilon^2)$ sparse vector $\mathbf{y} \in \mathbb{R}^n$ that satisfies (with constant probability that can be amplified as desired, see Section 2 for details):

$$\mathbf{y}^\top \mathbf{A} \mathbf{y} \geq \frac{1}{2}\mathcal{Z}^* - \epsilon \sqrt{\mathcal{Z}^*} \cdot \sqrt{\frac{\mathrm{Tr}(\mathbf{A})}{k}}.$$

Notice that the above bound depends on both $\mathcal{Z}^*$ and it square root and therefore is not a relative error bound. The second term scales as a function of the trace of $\mathbf{A}$ divided by $k$, which depends on the properties of the matrix $\mathbf{A}$ and the target sparsity.

The second algorithm is a deterministic thresholding scheme. It computes a small number of the top singular vectors of the matrix $\mathbf{A}$ and then applies a deterministic thresholding scheme on those singular vectors to (eventually) construct a sparse vector $\mathbf{z} \in \mathbb{R}^n$ that satisfies $\mathbf{z}^\top \mathbf{A} \mathbf{z} \geq (1/2)\mathcal{Z}^* - (3/2)\epsilon \, \mathrm{Tr}(\mathbf{A})$. Our analysis provides unconditional guarantees for the accuracy of the solution of this simple thresholding scheme. To the best of our knowledge, no such analyses have appeared in prior work (see Section 1.2 for details). The error bound of the second algorithm is weaker than the one provided by the first algorithm, but the second algorithm is deterministic and does not need to compute the square root (i.e., all singular vectors and singular values) of the matrix $\mathbf{A}$.

Our third algorithm provides novel bounds for the following standard convex relaxation of the problem of eqn. (1).

$$\max_{\mathbf{Z} \in \mathbb{R}^{n \times n}, \, \mathbf{Z} \succeq 0} \mathrm{Tr}(\mathbf{A}\mathbf{Z}) \text{ s.t. } \mathrm{Tr}(\mathbf{Z}) \leq 1 \text{ and } \sum |Z_{i,j}| \leq k. \tag{2}$$

It is well-known that the optimal solution to eqn. (2) is at least the optimal solution to eqn. (1). We present a novel, two-step rounding scheme that converts the optimal solution matrix $\mathbf{Z} \in \mathbb{R}^{n \times n}$ to a vector $\mathbf{z} \in \mathbb{R}^n$ that has expected sparsity[2] $\tilde{\mathcal{O}}(k^2/\epsilon^2)$ and satisfies $\mathbf{z}^\top \mathbf{A} \mathbf{z} \geq \gamma_Z(1 - \epsilon) \cdot \mathcal{Z}^* - \epsilon$. Here, $\gamma_Z$ is a constant that precisely depends on the top singular value of $\mathbf{Z}$, the condition number of $\mathbf{Z}$, and the extent to which the SDP relaxation of eqn. (2) is able to capture the original problem (see Theorem 4.1 and the following discussion for details). To the best of our knowledge, this is the first analysis of a rounding scheme for the convex relaxation of eqn. (2) that does not assume a specific model for the covariance matrix $\mathbf{A}$.

**Applications to Sparse Kernel PCA.** Our algorithms have immediate applications to sparse kernel PCA (SKPCA), where the input matrix $\mathbf{A} \in \mathbb{R}^{n \times n}$ is instead implicitly given as a kernel matrix whose entry $(i, j)$ is the value $k(i, j) := \langle \phi(\mathbf{X}_{i*}), \phi(\mathbf{X}_{j*}) \rangle$ for some kernel function $\phi$ that implicitly maps an observation vector into some high-dimensional feature space. Although $\mathbf{A}$ is not explicit,

---

[2]For simplicity of presentation and following the lines of (Fountoulakis et al., 2017), we assume that the rows and columns of the matrix $\mathbf{A}$ have unit norm; this assumption was not necessary for the previous two algorithms and can be removed as in (Fountoulakis et al., 2017). We are also hiding a poly-logarithmic factor for simplicity, hence the $\tilde{O}(\cdot)$ notation. See Theorem 4.1 for a detailed statement.

we can query all $\mathcal{O}\left(n^2\right)$ entries of $\mathbf{A}$ using $\mathcal{O}\left(n^2\right)$ time assuming an oracle that computes the kernel function $k$. We can then subsequently apply our SPCA algorithms and achieve polynomial time runtime with the same approximation guarantees.

## 1.2 PRIOR WORK

SPCA was formally introduced by (d'Aspremont et al., 2007); however, previously studied PCA approaches based on rotating (Jolliffe, 1995) or thresholding (Cadima & Jolliffe, 1995) the top singular vector of the input matrix seemed to work well, at least in practice, given sparsity constraints. Following (d'Aspremont et al., 2007), there has been an abundance of interest in SPCA. (Jolliffe et al., 2003) considered LASSO (SCoTLASS) on an $\ell_1$ relaxation of the problem, while (Zou & Hastie, 2005) considered a non-convex regression-type approximation, penalized similar to LASSO. Additional heuristics based on LASSO (Ando et al., 2009) and non-convex $\ell_1$ regularizations (Zou & Hastie, 2005; Zou et al., 2006; Sriperumbudur et al., 2007; Shen & Huang, 2008) have also been explored. Random sampling approaches based on non-convex $\ell_1$ relaxations (Fountoulakis et al., 2017) have also been studied; we highlight that unlike our approach, (Fountoulakis et al., 2017) solved a non-convex relaxation of the SPCA problem and thus perhaps relied on locally optimal solutions. Additionally, (Moghaddam et al., 2006b) considered a branch-and-bound heuristic motivated by greedy spectral ideas. (Journée et al., 2010; Papailiopoulos et al., 2013; Kuleshov, 2013; Yuan & Zhang, 2013) further explored other spectral approaches based on iterative methods similar to the power method. (Yuan & Zhang, 2013) specifically designed a sparse PCA algorithm with early stopping for the power method, based on the target sparsity.

Another line of work focused on using semidefinite programming (SDP) relaxations (d'Aspremont et al., 2007; d'Aspremont et al., 2008; Amini & Wainwright, 2009). Notably, (Amini & Wainwright, 2009) achieved provable theoretical guarantees regarding the SDP and thresholding approach of (d'Aspremont et al., 2007) in a *specific*, high-dimensional spiked covariance model, in which a base matrix is perturbed by adding a sparse maximal eigenvector. In other words, the input matrix is the identity matrix plus a "spike", i.e., a sparse rank-one matrix.

Despite the variety of heuristic-based sparse PCA approaches, very few theoretical guarantees have been provided for SPCA; this is partially explained by a line of hardness-of-approximation results. The sparse PCA problem is well-known to be NP-Hard (Moghaddam et al., 2006a). (Magdon-Ismail, 2017) shows that if the input matrix is not PSD, then even the *sign* of the optimal value cannot be determined in polynomial time unless P = NP, ruling out any multiplicative approximation algorithm. In the case where the input matrix is PSD, (Chan et al., 2016) shows that it is NP-hard to approximate the optimal value up to multiplicative $(1 + \epsilon)$ error, ruling out any polynomial-time approximation scheme (PTAS). Moreover, they show Small-Set Expansion hardness for any polynomial-time constant factor approximation algorithm and also that the standard SDP relaxation might have an exponential gap.

We conclude by summarizing prior work that offers provable guarantees (beyond the work of (Amini & Wainwright, 2009)), typically given *some assumptions about the input matrix*. (d'Aspremont et al., 2014) showed that the SDP relaxation can be used to find provable bounds when the covariance input matrix is formed by a number of data points sampled from Gaussian models with a single sparse singular vector. (Papailiopoulos et al., 2013) presented a combinatorial algorithm that analyzed a specific set of vectors in a low-dimensional eigenspace of the input matrix and presented relative error guarantees for the optimal objective, given the assumption that the input covariance matrix has a decaying spectrum. (Asteris et al., 2011) gave a polynomial-time algorithm that solves sparse PCA *exactly* for input matrices of constant rank. (Chan et al., 2016) showed that sparse PCA can be approximated in polynomial time within a factor of $n^{-1/3}$ and also highlighted an additive PTAS of (Asteris et al., 2015) based on the idea of finding multiple disjoint components and solving bipartite maximum weight matching problems. This PTAS needs time $n^{\mathrm{poly}(1/\epsilon)}$, whereas all of our algorithms have running times that are a low-degree polynomial in $n$.

## 2 SPARSE PCA VIA RANDOMIZED MATRIX MULTIPLICATION

Our first algorithm for SPCA leverages primitives and ideas from randomized matrix multiplication (Drineas & Kannan, 2001; Drineas et al., 2006; Drineas & Mahoney, 2016; 2018; Woodruff, 2014). Let $\mathbf{P} \in \mathbb{R}^{m \times n}$ and $\mathbf{Q} \in \mathbb{R}^{n \times p}$ and recall that their product $\mathbf{PQ}$ equals $\mathbf{PQ} = \sum_{i=1}^{n} \mathbf{P}_{*i}\mathbf{Q}_{i*}$.

Recall that $\mathbf{P}_{*i}$ denotes the $i$-th column of $\mathbf{P}$ and $\mathbf{Q}_{i*}$ the $i$-th row of $\mathbf{Q}$. A well-known approach to approximate the product $\mathbf{PQ}$ is to sample a subset of columns of $\mathbf{P}$ (we will do this without replacement) and the corresponding rows of $\mathbf{Q}$ (Drineas & Mahoney, 2018). Formally, let the random variables $Z_i \overset{iid}{\sim}$ Bernoulli($p_i$), $i = 1 \ldots n$, denote whether the $i$-th column of $\mathbf{P}$ and the $i$-th row of $\mathbf{Q}$ are sampled. Define the diagonal sampling-and-rescaling matrix $\mathbf{S} \in \mathbb{R}^{n \times n}$ by $\mathbf{S} \triangleq \text{diag}\left\{ Z_1/\sqrt{p_1}, \ldots, Z_n/\sqrt{p_n} \right\}$. The sampling probabilities $\{p_i\}_{i=1}^n$ *do not* have to sum up to one. The number of sampled column/row pairs, denoted by $s$, satisfies $\mathbb{E}[s] = \sum_{i=1}^n p_i$. (See Algorithm 1 for details.) The next lemma (see Appendix A.1 for its proof) presents accuracy bounds

---

**Algorithm 1** Construct sampling-and-rescaling matrix $\mathbf{S}$

**Input:** Probabilities $\tilde{p}_i$, $i = 1 \ldots n$ and integer $s \ll n$.
**Output:** Diagonal sampling-and-rescaling matrix $\mathbf{S} \in \mathbb{R}^{n \times n}$.
  1: **for** $i = 1$ **to** $n$ **do**
  2:     $\mathbf{S}_{ii} \leftarrow \begin{cases} 1/\sqrt{p_i}, & \text{with probability } p_i = \min\{s\tilde{p}_i, 1\} \\ 0, & \text{otherwise} \end{cases}$

---

when Algorithm 1 is used to approximate matrix multiplication.

**Lemma 2.1** *Given matrices $\mathbf{P} \in \mathbb{R}^{m \times n}$ and $\mathbf{Q} \in \mathbb{R}^{n \times p}$, let $\mathbf{S} \in \mathbb{R}^{n \times n}$ be constructed using Algorithm 1 with $\tilde{p}_i = \|\mathbf{P}_{*i}\|_2^2/\|\mathbf{P}\|_F^2$, for $i = 1 \ldots n$. Then,*

$$\mathbb{E}\left[ \|\mathbf{PS}^2\mathbf{Q} - \mathbf{PQ}\|_F^2 \right] \leq \frac{1}{s} \|\mathbf{P}\|_F^2 \|\mathbf{Q}\|_F^2 . \tag{3}$$

Our SPCA algorithm uses the above primitive to approximate the product of the (square root) of the input matrix $\mathbf{A}$ and its top right singular vector $\mathbf{v}$. Thus, the proposed SPCA algorithm *sparsifies* the top right singular vector of $\mathbf{v}$ of $\mathbf{A}$ without losing too much of the variance that is captured by $\mathbf{v}$. Interestingly, this conceptually simple algorithm has not been formally analyzed in prior work. Algorithm 2 details our approach.

---

**Algorithm 2** SPCA via randomized matrix multiplication

**Input:** $\mathbf{A} \in \mathbb{R}^{n \times n}$, sparsity parameter $k$, accuracy parameter $\epsilon \in (0, 1)$.
**Output:** $\mathbf{y} \in \mathbb{R}^n$ satisfying $\mathbb{E}[\|\mathbf{y}\|_2] \leq 1$ and $\mathbb{E}[\|\mathbf{y}\|_0] \leq k/\epsilon^2$.
  1. $\mathbf{X} \leftarrow \mathbf{A}^{1/2}$;
  2. Use Algorithm 1 to construct $\mathbf{S} \in \mathbb{R}^{n \times n}$ with $\tilde{p}_i = \|\mathbf{X}_{*i}\|_2^2/\|\mathbf{X}\|_F^2$ and $s = 4k/\epsilon^2$;
  3. Let $\mathbf{v} \in \mathbb{R}^n$ be the top right singular vector of $\mathbf{XS}$;
  4. $\mathbf{y} \leftarrow \mathbf{Sv}$;

---

**Theorem 2.2** *Let $k$ be the sparsity parameter and $\epsilon \in (0, 1]$ be the accuracy parameter. Let $\mathbf{S} \in \mathbb{R}^{n \times n}$ be the sampling matrix of Lemma 2.1 with $s = 4k/\epsilon^2$. Then, Algorithm 2 returns a vector $\mathbf{y}$ with expected sparsity at most $s$ (i.e., $\mathbb{E}[\|\mathbf{y}\|_0] \leq s$) and expected two norm at most one (i.e., $\mathbb{E}\left[\|\mathbf{y}\|_2^2\right] \leq 1$) such that with probability at least $1/4$, we have*

$$\mathbf{y}^\top \mathbf{A} \mathbf{y} \geq 1/2\mathcal{Z}^* - \epsilon \sqrt{\mathcal{Z}^*} \cdot \sqrt{\text{Tr}(\mathbf{A})/k}. \tag{4}$$

See Appendix A.1 for a proof of the above theorem. We note that the success probability of Algorithm 2 can be trivially amplified by repeating the algorithm $t$ times and keeping the vector $\mathbf{y}$ that maximizes $\mathbf{y}^\top \mathbf{A} \mathbf{y}$. Then, the failure probability of the overall approach diminishes exponentially fast as a function of $t$ to at most $(3/4)^t$. Finally, the running time of Algorithm 2 is dominated by the computation of a square root of the matrix $\mathbf{A}$ in the first step, which takes $\mathcal{O}(n^3)$ time via the computation of the SVD of $\mathbf{A}$.

## 3 SPCA VIA THRESHOLDING

Our second algorithm is based on a thresholding scheme using the top $\ell$ right singular vectors of the PSD matrix $\mathbf{A}$. Given $\mathbf{A}$ and an accuracy parameter $\epsilon$, our approach first computes $\mathbf{\Sigma}_\ell \in \mathbb{R}^{\ell \times \ell}$

(the diagonal matrix of the top $\ell$ singular values of $\mathbf{A}$) and $\mathbf{U}_\ell \in \mathbb{R}^{n \times \ell}$ (the matrix of the top $\ell$ right singular vectors of $\mathbf{A}$), for $\ell = 1/\epsilon$. Then, it *deterministically* selects a subset of $\mathcal{O}\left(k/\epsilon^2\right)$ columns of $\boldsymbol{\Sigma}_\ell^{1/2} \mathbf{U}_\ell^\top$ using a simple thresholding scheme based on the norms of the columns of $\boldsymbol{\Sigma}_\ell^{1/2} \mathbf{U}_\ell^\top$. (Recall that $k$ is the sparsity parameter of the SPCA problem.) In the last step, it returns the *top right singular vector* of the matrix consisting of the chosen columns of $\boldsymbol{\Sigma}_\ell^{1/2} \mathbf{U}_\ell^\top$. Notice that this right singular vector is an $\mathcal{O}\left(k/\epsilon^2\right)$-dimensional vector, which is finally expanded to a vector in $\mathbb{R}^n$ by appropriate padding with zeros. This sparse vector is our approximate solution to the SPCA problem of eqn. (1).

This simple algorithm is somewhat reminiscent of prior thresholding approaches for SPCA. However, to the best of our knowledge, no provable a priori bounds were known for such algorithms without strong assumptions on the input matrix. This might be due to the fact that prior approaches focused on thresholding only the top right singular vector of $\mathbf{A}$, whereas our approach thresholds the top $\ell = 1/\epsilon$ right singular vectors of $\mathbf{A}$. This slight relaxation allows us to present provable bounds for the proposed algorithm.

In more detail, let the SVD of $\mathbf{A}$ be $\mathbf{A} = \mathbf{U}\boldsymbol{\Sigma}\mathbf{U}^T$. Let $\boldsymbol{\Sigma}_\ell \in \mathbb{R}^{\ell \times \ell}$ be the diagonal matrix of the top $\ell$ singular values and let $\mathbf{U}_\ell \in \mathbb{R}^{n \times \ell}$ be the matrix of the top $\ell$ right (or left) singular vectors. Let $R = \{i_1, \ldots, i_{|R|}\}$ be the set of indices of rows of $\mathbf{U}_\ell$ that have squared norms at least $\epsilon/k$ and let $\bar{R}$ be its complement. Here $|R|$ denotes the cardinality of the set $R$ and $R \cup \bar{R} = \{1, \ldots, n\}$. Let $\mathbf{R} \in \mathbb{R}^{n \times |R|}$ be a sampling matrix that selects[3] the columns of $\mathbf{U}_\ell$ whose indices are in the set $R$. Given this notation, we are now ready to state Algorithm 3.

---

**Algorithm 3** SPCA via thresholding

---

**Input:** $\mathbf{A} \in \mathbb{R}^{n \times n}$, sparsity $k$, error parameter $\epsilon > 0$.
**Output:** $\mathbf{y} \in \mathbb{R}^n$ such that $\|y\|_2 = 1$ and $\|\mathbf{y}\|_0 = k/\epsilon^2$.
1: $\ell \leftarrow 1/\epsilon$;
2: Compute $\mathbf{U}_\ell \in \mathbb{R}^{n \times \ell}$ (top $\ell$ left singular vectors of $\mathbf{A}$) and $\boldsymbol{\Sigma}_\ell \in \mathbb{R}^{\ell \times \ell}$ (square roots of the top $\ell$ singular values of $\mathbf{A}$);
3: Let $R = \{i_1, \ldots, i_{|R|}\}$ be the set of rows of $\mathbf{U}_\ell$ with squared norms at least $\epsilon/k$ and let $\mathbf{R} \in \mathbb{R}^{n \times |R|}$ be the associated sampling matrix (see text for details);
4: $\mathbf{y} \in \mathbb{R}^{|R|} \leftarrow \operatorname{argmax}_{\|\mathbf{x}\|_2 = 1} \left\| \boldsymbol{\Sigma}_\ell \mathbf{U}_\ell^\top \mathbf{R}\mathbf{x} \right\|_2^2$;
5: **return** $\mathbf{z} = \mathbf{R}\mathbf{y} \in \mathbb{R}^n$;

---

Notice that $\mathbf{R}\mathbf{y}$ satisfies $\|\mathbf{R}\mathbf{y}\|_2 = \|\mathbf{y}\|_2 = 1$ (since $\mathbf{R}$ has orthogonal columns) and $\|\mathbf{R}\mathbf{y}\|_0 = |R|$. Since $R$ is the set of rows of $\mathbf{U}_\ell$ with squared norms at least $\epsilon/k$ and $\|\mathbf{U}_\ell\|_F^2 = \ell = 1/\epsilon$, it follows that $|R| \leq k/\epsilon^2$. Thus, the vector returned by Algorithm 3 has $k/\epsilon^2$ sparsity and unit norm.

**Theorem 3.1** *Let $k$ be the sparsity parameter and $\epsilon \in (0, 1]$ be the accuracy parameter. Then, the vector $\mathbf{z} \in \mathbb{R}^n$ (the output of Algorithm 3) has sparsity $k/\epsilon^2$, unit norm, and satisfies $\mathbf{z}^\top \mathbf{A}\mathbf{z} \geq (1/2)\mathcal{Z}^* - (3/2)\epsilon \operatorname{Tr}(\mathbf{A})$.*

We defer the proof of Theorem 3.1 to Appendix A.2. The running time of Algorithm 3 is dominated by the computation of the top $\ell$ singular vectors and singular values of the matrix $\mathbf{A}$. In practice, any iterative method, such as subspace iteration using a random initial subspace or the Krylov subspace of the matrix, can be used towards this end. However, our current analysis does not account for the inevitable approximation error incurred by such methods, which run in $\mathcal{O}\left(\operatorname{nnz}(\mathbf{A})\ell\right)$ time. One could always use the SVD of the full matrix $\mathbf{A}$ ($\mathcal{O}\left(n^3\right)$ time) to compute the top $\ell$ singular vectors and singular values of $\mathbf{A}$. Finally, we highlight that, as an intermediate step in the proof of Theorem 3.1, we need to prove the following lemma (see Appendix A.2 for its proof):

**Lemma 3.2** *Let $\mathbf{A} \in \mathbb{R}^{n \times n}$ be a PSD matrix and $\boldsymbol{\Sigma} \in \mathbb{R}^{n \times n}$ (respectively, $\boldsymbol{\Sigma}_\ell \in \mathbb{R}^{\ell \times \ell}$) be the diagonal matrix of all (respectively, top $\ell$) singular values and let $\mathbf{U} \in \mathbb{R}^{n \times n}$ (respectively, $\mathbf{U}_\ell \in \mathbb{R}^{n \times \ell}$) be the matrix of all (respectively, top $\ell$) singular vectors. Then, for all unit vectors $\mathbf{x} \in \mathbb{R}^n$,*

$$\left\| \boldsymbol{\Sigma}_\ell^{1/2} \mathbf{U}_\ell^\top \mathbf{x} \right\|_2^2 \geq \left\| \boldsymbol{\Sigma}^{1/2} \mathbf{U}^\top \mathbf{x} \right\|_2^2 - \epsilon \operatorname{Tr}(\mathbf{A}).$$

---

[3]Each column of $\mathbf{R}$ has a single non-zero entry (set to one), corresponding to one of the $|R|$ selected columns. Formally, $\mathbf{R}_{i_t, t} = 1$ for $t = 1, \ldots, |R|$; all other entries of $\mathbf{R}$ are set to zero.

The above simple lemma is very much at the heart of our proof of Theorem 3.1 and, unlike prior work, allows us to provide provably accurate bounds for the thresholding Algorithm 3.

**Using an approximate SVD solution.** The guarantees of Theorem 3.1 in Algorithm 3 uses an exact SVD computation, which could take time $\mathcal{O}\left(n^3\right)$. We can further improve the running time by using an approximate SVD algorithm such as the randomized block Krylov method of Musco & Musco (2015), which runs in nearly input sparsity runtime. Our analysis uses the relationships $\left\|\mathbf{\Sigma}_{\ell,\perp}^{1/2}\right\|_2^2 \leq \frac{\text{Tr}(\mathbf{A})}{\ell}$ and $\sigma_1(\Sigma_\ell) \leq \text{Tr}(\mathbf{A})$. The randomized block Krylov method of Musco & Musco (2015) recovers these guarantees up to a multiplicative $(1 + \epsilon)$, using $\mathcal{O}\left(\frac{\log n}{\epsilon^2} \cdot \text{nnz}(\mathbf{A})\right)$ runtime. Thus by rescaling $\epsilon$, we recover the same guarantees of Theorem 3.1 by using an approximate SVD with nearly input sparsity runtime.

## 4 SPCA VIA A SEMIDEFINITE PROGRAMMING RELAXATION

Our third algorithm is based on the SDP relaxation of eqn. (2). Recall that solving eqn. (2) returns a PSD matrix $\mathbf{Z}^* \in \mathbb{R}^{n \times n}$ that, by the definition of the semidefinite programming relaxation, satisfies $\text{Tr}(\mathbf{A}\mathbf{Z}^*) \geq \mathcal{Z}^*$, where $\mathcal{Z}^*$ is the true optimal solution of SPCA in eqn. (2).

We now need to convert the matrix $\mathbf{Z}^* \in \mathbb{R}^{n \times n}$ into a sparse vector that will be the output of our approximate SPCA algorithm *and* will satisfy certain accuracy guarantees. Towards that end, we employ a novel two-step rounding procedure. First, a critical observation is that generating a random Gaussian vector $\mathbf{g} \in \mathbb{R}^n$ and computing the vector $\mathbf{Z}^*\mathbf{g} \in \mathbb{R}^n$ results in an unbiased estimator for the trace of $(\mathbf{Z}^*)^\top \mathbf{A}\mathbf{Z}^*$ in the following sense: $\mathbb{E}\left[\mathbf{g}^\top (\mathbf{Z}^*)^\top \mathbf{A}\mathbf{Z}^*\mathbf{g}\right] = \text{Tr}((\mathbf{Z}^*)^\top \mathbf{A}\mathbf{Z}^*)$.

Using von Neumann's trace inequality, we can prove that

$$\mathbb{E}\left[\mathbf{g}^\top (\mathbf{Z}^*)^\top \mathbf{A}\mathbf{Z}^*\mathbf{g}\right] = \text{Tr}((\mathbf{Z}^*)^\top \mathbf{A}\mathbf{Z}^*) \geq \gamma_{Z^*} \cdot \text{Tr}(\mathbf{A}\mathbf{Z}^*) \geq \gamma_Z \cdot \mathcal{Z}^*. \tag{5}$$

Here $\gamma_{Z^*}$ is a constant that precisely depends on the top singular value of $\mathbf{Z}^*$, the condition number of $\mathbf{Z}^*$, and the extent to which the SDP relaxation of eqn. (2) is able to capture the original problem (see Theorem 4.1 for the exact expression of $\gamma_{Z^*}$). The above inequality implies that, at least in expectation, we could use the vector $\mathbf{Z}^*\mathbf{g}$ as a "rounding" of the output of the semidefinite programming relaxation. However, there is absolutely no guarantee that the vector $\mathbf{Z}^*\mathbf{g}$ is sparse. Thus, in order to sparsify $\mathbf{Z}^*\mathbf{g}$, we employ a separate sparsification procedure, where each entry of $\mathbf{Z}^*\mathbf{g}$ is kept (and rescaled) with probability proportional to its magnitude. This procedure is similar to the one proposed in (Fountoulakis et al., 2017) and guarantees that larger entries of $\mathbf{Z}^*\mathbf{g}$ are more likely to be kept, while smaller entries of $\mathbf{Z}^*\mathbf{g}$ are more likely to be set to zero, without too much loss in accuracy. We also note that to ensure a sufficiently high probability of success for the overall approach, we generate multiple Gaussian vectors and keep the one that maximizes the quantity $\mathbf{g}^\top (\mathbf{Z}^*)^\top \mathbf{A}\mathbf{Z}^*\mathbf{g}$. See Algorithm 4 and Algorithm 5 for a detailed presentation of our approach.

---

**Algorithm 4** SPARSIFY

---

**Input:** $\mathbf{y} \in \mathbb{R}^n$ and sparsity parameter $s$.
**Output:** $\mathbf{z} \in \mathbb{R}^n$ with $\mathbb{E}\left[\|\mathbf{z}\|_0\right] \leq s$.
 1: **for** $i = 1 \ldots n$ **do**
 2: $\quad \mathbf{z}_i := \begin{cases} \frac{1}{p_i}\mathbf{y}_i, & \text{with probability } p_i = \min\left\{1, \frac{s|y_i|}{\|\mathbf{y}\|_1}\right\}, \\ 0 & \text{otherwise.} \end{cases}$

---

The running time of the algorithm is dominated by the time needed to solve the semidefinite programming relaxation of eqn. (2), which, in our setting, is $\mathcal{O}\left(n^{3.5}\right)$ (Alizadeh, 1995). We do note that SDP solvers such as the one in (Alizadeh, 1995) return an additive error approximation to the optimal solution. However, the running time dependence of SDP solvers on the additive error $\gamma$ is logarithimic in $1/\gamma$ and thus highly accurate approximations can be derived without a significant increase in the number of iterations of the solver. Thus, for the sake of clarity, we initially omit this additive error from the analysis and address the approximate solution at the end.

Our main quality-of-approximation result for Algorithm 5 is Theorem 4.1. For simplicity of presentation, and following the lines of (Fountoulakis et al., 2017), we assume that all rows and columns

---

**Algorithm 5** Rounding-based SPCA

---

**Input:** PSD matrix $\mathbf{A} \in \mathbb{R}^{n \times n}$, error tolerance $\epsilon > 0$, and sparsity parameter $k$.
**Output:** $\mathbf{x} \in \mathbb{R}^n$ with $\mathbb{E}\left[\|\mathbf{x}\|_0\right] = s$.
1: Let $\mathbf{Z}^*$ be the optimal solution to the relaxed SPCA problem of eqn. (2);
2: $M \leftarrow {}^{80}/\epsilon^2$ and $s \leftarrow O\left(\frac{k^2 \log^{5/2}(1/\epsilon)}{\epsilon^2}\right)$;            ▷See Theorem A.13 for the exact value of $s$.
3: Generate $M$ random Gaussian vectors $g_1, \ldots, g_M$ in $\mathbb{R}^n$;
4: $\mathbf{y} \leftarrow \mathbf{Z}^* g_j$, where $j \leftarrow \operatorname{argmax}_{i=1 \ldots M} \mathbf{g}_i^\top (\mathbf{Z}^*)^\top \mathbf{A}\mathbf{Z}^* \mathbf{g}_i$;
5: $\mathbf{z} \leftarrow \text{SPARSIFY}(\mathbf{y}, s)$;

---

of $\mathbf{A}$ have been normalized to have unit norms. This assumption can be relaxed as in (Fountoulakis et al., 2017). In the statement of the theorem, we will use the notation $\mathbf{Z}_1$ to denote the best rank-one approximation to the matrix $\mathbf{Z}$.

**Theorem 4.1** *Given a PSD matrix $\mathbf{A} \in \mathbb{R}^{n \times n}$, a sparsity parameter $k$, and an error tolerance $\epsilon > 0$, let $\mathbf{Z}$ be an optimal solution to the relaxed SPCA problem of eqn. (2). Assume that*

$$\operatorname{Tr}(\mathbf{A}\mathbf{Z}) \leq \alpha \operatorname{Tr}(\mathbf{A}\mathbf{Z}_1) \tag{6}$$

*for some constant $\alpha \geq 1$. Then, Algorithm 5 outputs a vector $\mathbf{z} \in \mathbb{R}^n$ that, with probability at least $5/8$, satisfies $\mathbb{E}\left[\|\mathbf{z}\|_0\right] = \mathcal{O}\left(k^2 \log^{5/2}(1/\epsilon)/\epsilon^2\right)$, $\|\mathbf{z}\|_2 = \mathcal{O}\left(\sqrt{\log^{1/\epsilon}}\right)$, and*

$$\mathbf{z}^\top \mathbf{A} \mathbf{z} \geq \gamma_Z (1 - \epsilon) \cdot \mathcal{Z}^* - \epsilon.$$

*Here $\gamma_Z = \left(1 - \left(1 - \frac{1}{\kappa(\mathbf{Z})}\right)\left(1 - \frac{1}{\alpha}\right)\right) \sigma_1(\mathbf{Z})$ with $\sigma_1(\mathbf{Z})$ and $\kappa(\mathbf{Z})$ being the top singular value and condition number of $\mathbf{Z}$ respectively.*

Similar to Theorem 2.2, the probability of success can be boosted to $1 - \delta$ by repeating the algorithm $\mathcal{O}\left(\frac{1}{\delta}\right)$ times in parallel. Moreover by using Markov's inequality, we can also guarantee a vector $\mathbf{z}$ with sparsity $\mathcal{O}\left(k^2 \log^{5/2}(1/\epsilon)/\epsilon^2\right)$ with probability $1 - \delta$, rather than just in expectation.

We now discuss the condition of eqn. (6) and the constant $\gamma_Z$. Our assumption simply says that much of the trace of the matrix $\mathbf{A}\mathbf{Z}$ should be captured by the trace of $\mathbf{A}\mathbf{Z}_1$, as quantified by the constant $\alpha$. For example, if $\mathbf{Z}$ were a rank-one matrix, then the assumption would hold with $\alpha = 1$. As the trace of $\mathbf{A}\mathbf{Z}_1$ fails to approximate the trace of $\mathbf{A}\mathbf{Z}$ (which intuitively implies that the SDP relaxation of eqn. (2) did not sufficiently capture the original problem) the constant $\alpha$ increases and the quality of the approximation decreases. More precisely, first notice that the constant $\gamma_Z$ is upper bounded by one, because $\sigma_1(\mathbf{Z}) \leq 1$ by the SDP relaxation. Second, the quality of the approximation increases as $\gamma_Z$ approaches one. This happens if either the condition number of $\mathbf{Z}$ is close to one *or* if the constant $\alpha$ is close to one; at the same time, $\sigma_1(\mathbf{Z})$ also needs to be close to one. Clearly, these conditions are satisfied if $\mathbf{Z}$ is well approximated by $\mathbf{Z}_1$. In our experiments, we indeed observed that $\alpha$ is close to one and that the top singular value of $\mathbf{Z}$ is close to one, which imply that $\gamma_Z$ is also close to one (Appendix, Table 6). The proof of Theorem 4.1 is delegated to Appendix A.3 (as Theorem A.13), but we outline here a summary of statements that lead to the final bound.

**Lemma 4.2** *Let $\mathbf{y}$ and $\mathbf{z}$ be defined as in Algorithm 5. If $\|\mathbf{y}\|_1 \leq \alpha$ and $\|\mathbf{y}\|_2 \leq \beta$, then*

$$|\mathbf{y}^\top \mathbf{A}\mathbf{y} - \mathbf{z}^\top \mathbf{A}\mathbf{z}| \leq 2|\mathbf{y}^\top \mathbf{A}(\mathbf{y} - \mathbf{z})| + |(\mathbf{y} - \mathbf{z})^\top \mathbf{A}(\mathbf{y} - \mathbf{z})|.$$

*Moreover, with probability at least $7/8$, we have $|\mathbf{y}^\top \mathbf{A}(\mathbf{y} - \mathbf{z})| \leq {}^{4\alpha\beta}/\sqrt{s}$ and $|(\mathbf{y} - \mathbf{z})^\top \mathbf{A}(\mathbf{y} - \mathbf{z})| \leq \sqrt{\frac{64\alpha^4}{s^2} + \frac{96\alpha\beta^3}{s}}.$*

**Lemma 4.3** *Let $M = {}^{80}/\epsilon^2$, $\alpha = k(1 + 2\sqrt{\log M})$, and $\beta = 2\sqrt{\log M}$. If the sparsity parameter $s$ is set to $s = {}^{450\alpha^2\beta^3}/\epsilon^2$, then with probability at least $3/4$, we have $\mathbf{y}^\top \mathbf{A}\mathbf{y} \leq \mathbf{z}^\top \mathbf{A}\mathbf{z} + \epsilon.$*

Letting $\mathbf{y} = \mathbf{Z}^* \mathbf{g}$, we now conclude the proof by combining eqn. (5) with the above lemma to bound (at least in expectation) the accuracy of Algorithm 5. To get a high probability bound, we leverage a result by (Avron & Toledo, 2011) on estimating the trace of PSD matrices. This approach allows us

to properly analyze step 4 of Algorithm 5, which uses multiple random Gaussian vectors to achieve measure concentration (see Appendix A.3 for details).

Finally, we can bound the $\ell_2$ norm of the vector $\mathbf{z}$ of Algorithm 5 by proving that, with probability at least $3/4$, $\|\mathbf{z}\|_2 = \mathcal{O}\left(\sqrt{\log 1/\epsilon}\right)$. Notice that this slightly relaxes the requirement that $\mathbf{z}$ has unit norm; however, even for accuracy $\epsilon$ close to machine precision, $\sqrt{\log 1/\epsilon}$ is a small constant.

**Using approximate SDP solution.** The guarantees of Theorem 4.1 in Algorithm 5 uses an optimal solution $\mathbf{Z}^*$ to the SDP relaxation in eqn. (2). In practice, we will only obtain an approximate solution $\tilde{\mathbf{Z}}$ to eqn. (2) using any standard SDP solver, e.g. (Alizadeh, 1995), such that $\text{Tr}(\mathbf{A}\tilde{\mathbf{Z}}) \geq \text{Tr}(\mathbf{A}\mathbf{Z}^*) - \epsilon$ after $\mathcal{O}\left(\log\frac{1}{\epsilon}\right)$ iterations. Since our analysis only uses the relationship $(\mathbf{x}^*)^\top \mathbf{A}\mathbf{x}^* \leq \text{Tr}(\mathbf{A}\mathbf{Z}^*)$, then the additive $\epsilon$ guarantee can be absorbed into the other $\epsilon$ factors in the guarantees of Theorem 4.1. Thus, we recover the same guarantees of Theorem 4.1 by using an approximate solution to the SDP relaxation in eqn. (2).

## 5 EXPERIMENTS

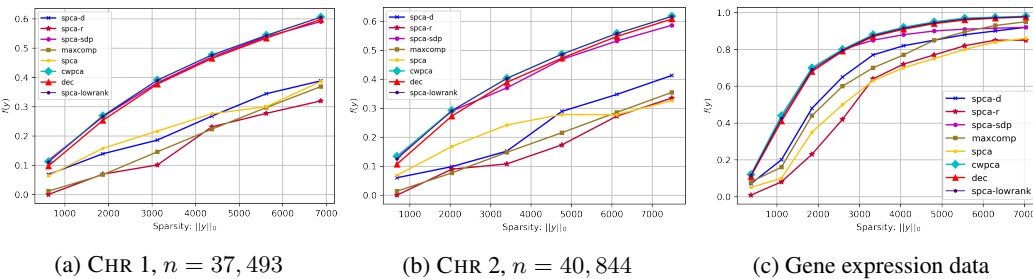

(a) CHR 1, $n = 37,493$    (b) CHR 2, $n = 40,844$    (c) Gene expression data

Fig. 1: Experimental results on real data: $f(\mathbf{y})$ vs. sparsity.

| | topdiam | length | moist | testsg | ovensg | ringtop | ringbut | bowmax | bowdist | whorls | clear | knots | diaknot | PVE | $\mathcal{Z}^*$ |
|---|---|---|---|---|---|---|---|---|---|---|---|---|---|---|---|
| spca-d (Algorithm 3) | −0.420 | −0.422 | 0 | 0 | 0 | −0.296 | −0.416 | −0.305 | −0.371 | −0.394 | 0 | 0 | 0 | 30.71% | 3.993 |
| spca-r (Algorithm 2) | −0.448 | −0.445 | 0 | 0 | 0 | −0.379 | −0.212 | −0.179 | −0.398 | −0.471 | 0 | 0 | 0 | 28.65% | 3.725 |
| spca-sdp (Algorithm 5) | −0.424 | −0.430 | 0 | 0 | 0 | −0.268 | −0.403 | −0.313 | −0.379 | −0.399 | 0 | 0 | 0 | 30.74% | 3.996 |
| dec (Yuan et al., 2019) | −0.423 | −0.430 | 0 | 0 | 0 | −0.268 | −0.403 | −0.313 | −0.379 | −0.399 | 0 | 0 | 0 | 30.74% | 3.996 |
| cwpca (Beck & Vaisbourd, 2016) | −0.423 | −0.430 | 0 | 0 | 0 | −0.268 | −0.403 | −0.313 | −0.379 | −0.399 | 0 | 0 | 0 | 30.74% | 3.996 |
| spca-lowrank (Papailiopoulos et al., 2013) | −0.427 | −0.432 | 0 | 0 | 0 | −0.249 | −0.390 | −0.326 | −0.383 | −0.403 | 0 | 0 | 0 | 30.72% | 3.994 |
| spca (Zou et al., 2006) | −0.477 | −0.476 | 0 | 0 | 0.177 | 0 | −0.250 | −0.344 | −0.416 | −0.400 | 0 | 0 | 0 | 28.01% | 3.641 |
| dspca (d'Aspremont et al., 2007) | −0.491 | −0.507 | 0 | 0 | 0 | −0.067 | −0.357 | −0.234 | −0.387 | −0.409 | 0 | 0 | 0 | 29.39% | 3.821 |

Table 1: Loadings, % of variance explained (PVE), and the objective function value for the first pc of the Pit Props data.

We compare the outputs our algorithms with that of the state-of-the-art sparse PCA solvers such as the coordinate-wise optimization algorithm of Beck & Vaisbourd (2016) (cwpca) and the block decomposition algorithm of Yuan et al. (2019) (dec), along with other standard methods such as Papailiopoulos et al. (2013) (spca-lowrank), d'Aspremont et al. (2007) (dspca), and Zou et al. (2006) (spca). For the implementation of dec, we use *coordinate descent method* with workset: $(6, 6)$ and for cwpca, we use *greedy coordinate wise* (GCW) method.

First, in order to explore the sparsity patterns of the outputs of our algorithms and how close they are as compared to standard methods, we first apply our methods on the *pit props* data which was introduced in (Jeffers, 1967) and is a benchmark example used to test sparse PCA. It is a $13 \times 13$ correlation matrix, originally calculated from 180 observations with 13 explanatory variables. While the existing algorithms aimed to extract top 6 principal components, we restrict only to the top principal component. In particular, we are interested to apply our algorithms on the *Pit Props* matrix in a view to extract the top pc having a sparsity pattern similar to that of Beck & Vaisbourd (2016), Yuan et al. (2019), (Zou et al., 2006), Papailiopoulos et al. (2013), and (d'Aspremont et al., 2007). It is known that the decomposition method of Yuan et al. (2019) can find the global optimal solution. We take $k = 7$ and Table 1 shows that while spca-d (Algorithm 3) and spca-r (Algorithm 2) perform very similar to spca or that of dspca with % of variace explained (PVE) uniformly better than spca, our SDP-based method spca-sdp (Algorithm 5) exactly recovers the optimal solution and the output matches with both dec and cwpca and very close to spca-lowrank. We also

apply our algorithms on another benchmark artificial example from (Zou et al., 2006), please see Appendix B.3 for a detailed discussion.

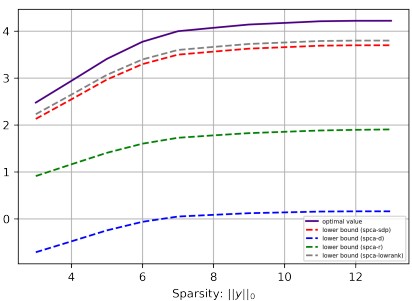

**Tightness of our bounds.** Now, we also verify the tightness of the theoretical lower bounds of our results with the guarantee of (Papailiopoulos et al., 2013) on the *pit props* data. We take $\epsilon = 0.1$ and found that the lower bound of our spca-sdp (Algorithm 5) (dashed red line on the left) is indeed very close to that of Papailiopoulos et al. (2013) with d=3 (dashed grey line on the left) . Nevertheless, the accuracy parameter of Papailiopoulos et al. (2013) typically relies on the spectrum of $\mathbf{A}$ *i.e.,* for a highly accurate output, $\epsilon$ can be much smaller depending on the structure of $\mathbf{A}$, in which case the difference between the lower bounds of our Algorithm 5 and Papailiopoulos et al. (2013) becomes even smaller.

Next, we further demonstrate the empirical performance of our algorithms on larger real-world datasets as well as on a synthetic dataset, similar to (Fountoulakis et al., 2017) (see Appendix B). We use human genetic data from the HGDP (Consortium, 2007) and HAPMAP (Li et al., 2008) (22 matrices, one for each chromosome). In addition, we also use a lung cancer gene expression dataset $(107 \times 22, 215)$ from (Landi et al., 2008) and a sparse document-term matrix $(2, 858 \times 12, 427)$ created using the Text-to-Matrix Generator (TMG) (Zeimpekis & Gallopoulos, 2006) (see Appendix B).

**Comparisons and metrics.** We compare our Algorithm 2 (spca-r), Algorithm 3 (spca-d), and Algorithm 5 (spca-sdp) with the solutions returned by spca (Zou et al., 2006), as well as the simple MaxComp heuristic (Cadima & Jolliffe, 1995). We define the quantity $f(\mathbf{y}) = \mathbf{y}^\top \mathbf{A} \mathbf{y} / \|\mathbf{A}\|_2$ to measure the quality of an approximate solution $\mathbf{y} \in \mathbb{R}^n$ to the SPCA problem. Notice that $0 \leq f(\mathbf{y}) \leq 1$ for all $\mathbf{y}$ with $\|\mathbf{y}\|_2 \leq 1$. As $f(\mathbf{y})$ gets closer to one, the vector $\mathbf{y}$ captures more of the variance of the matrix $\mathbf{A}$ that corresponds to its top singular value and corresponding singular vector. Our goal is to identify a sparse vector $\mathbf{y}$ with $f(\mathbf{y}) \approx 1$. Since the outputs of our Algorithm 2 and Algorithm 5 may have norms that slightly exceed one and in order to have a fair comparison between different methods, we normalize our outputs in the same way as in (Fountoulakis et al., 2017) (Appendix B).

**Results.** In our experiments, for spca-d, spca-r, and spca-sdp, we fix the sparsity $s$ to be equal to $k$, so that all algorithms return the same number of non-zero elements. In Figures 1a-1c we evaluate the performance of the different SPCA algorithms by plotting $f(\mathbf{y})$ against $\|\mathbf{y}\|_0$, *i.e.,* the sparsity of the output vector, on data from chromosome 1, chromosome 2, and the gene expression data. Note that performance of our SDP-based method (spca-sdp) is indeed comparable with the *state-of-the-art* dec, cwpca, and spca-lowrank, while both spca-d and spca-r are *better than or at least comparable to* both spca and maxcomp. However, in practice, the running time of the SDP relaxation is substantially higher than our other methods, which highlights the interesting trade-offs between the accuracy and computation discussed in (Amini & Wainwright, 2009). See Apeendix B for more experimental results.

## 6 CONCLUSION AND OPEN PROBLEMS

We present three provably accurate, polynomial time, approximation algorithms for SPCA, without imposing restrictive assumptions on the input covariance matrix. Future directions include: *(i)* extend the proposed algorithms to handle more than one sparse singular vector by deflation or other strategies and *(ii)* explore matching lower bounds and/or improve the guarantees of Theorems 2.2, 3.1, and 4.1.

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

# Appendix to
# Approximation Algorithms for Sparse Principal
# Component Analysis

## A  APPENDIX

### A.1  SPCA VIA RANDOMIZED MATRIX MULTIPLICATION: PROOFS

First, we prove two lemmas that are crucial in proving Lemma 2.1.

**Lemma A.1** *Given matrices* $\mathbf{P} \in \mathbb{R}^{m \times n}$ *and* $\mathbf{Q} \in \mathbb{R}^{n \times p}$*, let* $\mathbf{S} \in \mathbb{R}^{n \times n}$ *be constructed using Algorithm 1. Then,*

$$\mathbb{E}\left[(\mathbf{PS}^2\mathbf{Q})_{ij}\right] = (\mathbf{PQ})_{ij}$$
$$\mathrm{var}\,(\mathbf{PS}^2\mathbf{Q})_{ij} = \sum_{k=1}^{n} \frac{\mathbf{P}_{ik}^2 \mathbf{Q}_{kj}^2}{p_k} - \sum_{k=1}^{n} \mathbf{P}_{ik}^2 \mathbf{Q}_{kj}^2.$$

*for any indices* $i, j \in \{1, \dots, n\}$*.*

**Proof :**  For any $i, j \in \{1, \dots, n\}$, we have that

$$\mathbb{E}\left[(\mathbf{PS}^2\mathbf{Q})_{ij}\right] = \mathbb{E}\left[\sum_{k=1}^{n} \mathbf{P}_{ik} \mathbf{S}_{kk}^2 \mathbf{Q}_{kj}\right] = \mathbb{E}\left[\sum_{k=1}^{n} \mathbf{P}_{ik} \left(\frac{Z_k^2}{p_k}\right) \mathbf{Q}_{kj}\right] = \sum_{k=1}^{n} \left(\frac{\mathbf{P}_{ik}\mathbf{Q}_{kj}}{p_k}\right) \mathbb{E}\left[Z_k^2\right]$$
$$= \sum_{k=1}^{n} \mathbf{P}_{ik}\mathbf{Q}_{kj} = (\mathbf{PQ})_{ij}\,,$$

since $Z_k^2 \overset{d}{=} Z_k \sim \mathrm{Ber}(p)$ and thus $\mathbb{E}\left[Z_k^2\right] = \mathbb{E}\left[Z_k\right] = p_k$, where $\overset{d}{=}$ denotes equality in distribution. By the independence of the $Z_k$'s, and noting that $\mathrm{var}\,(Z_k) = p_k(1 - p_k)$, we have that

$$\mathrm{var}\,(\mathbf{PS}^2\mathbf{Q})_{ij} = \mathrm{var} \sum_{k=1}^{n} \mathbf{P}_{ik} \left(\frac{Z_k^2}{p_k}\right) \mathbf{Q}_{kj}$$
$$= \sum_{k=1}^{n} \left(\frac{\mathbf{P}_{ik}\mathbf{Q}_{kj}}{p_k}\right)^2 \mathrm{var}\, Z_k^2 = \sum_{k=1}^{n} \left(\frac{1 - p_k}{p_k}\right) \mathbf{P}_{ik}^2 \mathbf{Q}_{kj}^2$$
$$= \sum_{k=1}^{n} \frac{\mathbf{P}_{ik}^2 \mathbf{Q}_{kj}^2}{p_k} - \sum_{k=1}^{n} \mathbf{P}_{ik}^2 \mathbf{Q}_{kj}^2\,.$$

$\square$

**Lemma A.2** *Given matrices* $\mathbf{P} \in \mathbb{R}^{m \times n}$ *and* $\mathbf{Q} \in \mathbb{R}^{n \times p}$*, let* $\mathbf{S} \in \mathbb{R}^{n \times n}$ *be constructed using Algorithm 1. Then,*

$$\mathbb{E}\left[\|\mathbf{PS}^2\mathbf{Q} - \mathbf{PQ}\|_F^2\right] = \sum_{i=1}^{n} \frac{\|\mathbf{P}_{*i}\|_2^2 \cdot \|\mathbf{Q}_{i*}\|_2^2}{p_i} - \sum_{i=1}^{n} \|\mathbf{P}_{*i}\|_2^2 \|\mathbf{Q}_{i*}\|_2^2. \tag{7}$$

*Here* $\mathbf{P}_{*i}$ *and* $\mathbf{Q}_{i*}$ *are the* $i$*-th column of* $\mathbf{P}$ *and* $i$*-th row of* $\mathbf{Q}$ *respectively.*

**Proof :**  Using Lemma A.1, we have that

$$\mathbb{E}\left[\|\mathbf{PQ} - \mathbf{PS}^2\mathbf{Q}\|_F^2\right] = \sum_{i=1}^{m}\sum_{j=1}^{p} \mathbb{E}\left[\left((\mathbf{PQ})_{ij} - (\mathbf{PS}^2\mathbf{Q})_{ij}\right)^2\right] = \sum_{i=1}^{m}\sum_{j=1}^{p} \mathrm{var}\,(\mathbf{PS}^2\mathbf{Q})_{ij}$$
$$= \sum_{i=1}^{m}\sum_{j=1}^{p} \left[\sum_{k=1}^{n} \frac{\mathbf{P}_{ik}^2 \mathbf{Q}_{kj}^2}{p_k} - \sum_{k=1}^{n} \mathbf{P}_{ik}^2 \mathbf{Q}_{kj}^2\right]$$

$$= \sum_{k=1}^{n} \left( \frac{1}{p_k} - 1 \right) \left( \sum_{i=1}^{m} \mathbf{P}_{ik}^2 \right) \left( \sum_{j=1}^{p} \mathbf{Q}_{kj}^2 \right)$$

$$= \sum_{k=1}^{n} \frac{\|\mathbf{P}_{*k}\|_2^2 \|\mathbf{Q}_{k*}\|_2^2}{p_k} - \sum_{k=1}^{n} \|\mathbf{P}_{*k}\|_2^2 \|\mathbf{Q}_{k*}\|_2^2 .$$

$\square$

**Proof of Lemma 2.1:** From Lemma A.2,

$$\mathbb{E}\left[\|\mathbf{P}\mathbf{S}^2\mathbf{Q} - \mathbf{P}\mathbf{Q}\|_F^2\right] = \sum_{i=1}^{n} \frac{\|\mathbf{P}_{*i}\|_2^2 \cdot \|\mathbf{Q}_{i*}\|_2^2}{p_i} - \sum_{i=1}^{n} \|\mathbf{P}_{*i}\|_2^2 \|\mathbf{Q}_{i*}\|_2^2$$

$$= \sum_{\{i:\tilde{p}_i \leq 1/s\}} \frac{\|\mathbf{P}_{*i}\|_2^2 \cdot \|\mathbf{Q}_{i*}\|_2^2}{s\tilde{p}_i} + \sum_{\{i:\tilde{p}_i > 1/s\}} \|\mathbf{P}_{*i}\|_2^2 \cdot \|\mathbf{Q}_{i*}\|_2^2 - \sum_{i=1}^{n} \|\mathbf{P}_{*i}\|_2^2 \|\mathbf{Q}_{i*}\|_2^2$$

$$\leq \sum_{\{i:\tilde{p}_i \leq 1/s\}} \frac{\|\mathbf{P}_{*i}\|_2^2 \cdot \|\mathbf{Q}_{i*}\|_2^2}{s\tilde{p}_i} \leq \sum_{i=1}^{n} \frac{\|\mathbf{P}_{*i}\|_2^2 \cdot \|\mathbf{Q}_{i*}\|_2^2}{s\tilde{p}_i} .$$

We conclude the proof by setting $\tilde{p}_i = \|\mathbf{P}_{*i}\|_2^2 / \|\mathbf{P}\|_F^2$. $\blacksquare$

**Proof of Theorem 2.2. Proof :** In Lemma 2.1, let $\mathbf{P} = \mathbf{X}$ and $\mathbf{Q} = \mathbf{x}^*$ to get

$$\mathbb{E}\left[\|\mathbf{X}\mathbf{S}^2\mathbf{x}^* - \mathbf{X}\mathbf{x}^*\|_2^2\right] \leq \frac{1}{s}\|\mathbf{X}\|_F^2 \cdot \|\mathbf{x}^*\|_2^2 \leq \frac{1}{s}\|\mathbf{X}\|_F^2 . \tag{8}$$

The last inequality follows from $\|\mathbf{x}^*\|_2 \leq 1$. Moreover, by Markov's inequality, with probability at least $3/4$,

$$\|\mathbf{X}\mathbf{S}^2\mathbf{x}^* - \mathbf{X}\mathbf{x}^*\|_2^2 \leq \frac{4}{s}\|\mathbf{X}\|_F^2 = \frac{\epsilon^2}{k}\|\mathbf{X}\|_F^2. \tag{9}$$

Let $\tilde{\mathbf{x}} = \mathbf{S}\mathbf{x}^*$. Taking square roots of both sides of the above inequality, and applying the triangle inequality on the left hand side of the above inequality, we get:

$$|\|\mathbf{X}\mathbf{x}^*\|_2 - \|\mathbf{X}\mathbf{S}\tilde{\mathbf{x}}\|_2| \leq \frac{\epsilon}{\sqrt{k}}\|\mathbf{X}\|_F$$

$$\Rightarrow \|\mathbf{X}\mathbf{S}\tilde{\mathbf{x}}\|_2 \geq \sqrt{\mathcal{Z}^*} - \frac{\epsilon}{\sqrt{k}}\|\mathbf{X}\|_F$$

$$\Rightarrow \|\mathbf{X}\mathbf{S}\tilde{\mathbf{x}}\|_2^2 \geq \mathcal{Z}^* + \frac{\epsilon^2}{k}\|\mathbf{X}\|_F^2 - \frac{2\epsilon}{\sqrt{k}}\sqrt{\mathcal{Z}^*} \cdot \|\mathbf{X}\|_F . \tag{10}$$

Note that $\mathcal{Z}^* = \|\mathbf{X}\mathbf{x}^*\|_2^2$. Ignoring the non-negative term in eqn. (10), we conclude

$$\|\mathbf{X}\mathbf{S}\tilde{\mathbf{x}}\|_2^2 \geq \mathcal{Z}^* - \frac{2\epsilon}{\sqrt{k}}\sqrt{\mathcal{Z}^*} \cdot \|\mathbf{X}\|_F. \tag{11}$$

Next, using sub-multiplicativity on the left hand side of eqn. (11),

$$\|\mathbf{X}\mathbf{S}\tilde{\mathbf{x}}\|_2^2 \leq \|\mathbf{X}\mathbf{S}\|_2^2\|\tilde{\mathbf{x}}\|_2^2 = \|\mathbf{X}\mathbf{S}\mathbf{v}\|_2^2\|\tilde{\mathbf{x}}\|_2^2 , \tag{12}$$

where $\mathbf{v} \in \mathbb{R}^n$ is the top right singular vector of $\mathbf{X}\mathbf{S}$. Letting $\mathbf{x}_i^*$ be the $i$-th entry of $\mathbf{x}^*$, we have,

$$\mathbb{E}(\|\tilde{\mathbf{x}}\|_2^2) = \mathbb{E}\left(\sum_{i=1}^{n} \frac{\mathbf{x}_i^{*2} Z_i^2}{p_i}\right) = \sum_{i=1}^{n} \frac{\mathbf{x}_i^{*2} p_i}{p_i} = \|\mathbf{x}^*\|_2^2 = 1 , \tag{13}$$

since $\mathbb{E}\left[Z_i^2\right] = p_i$. Using Markov's inequality, with probability at least $1/2$,

$$\|\tilde{\mathbf{x}}\|_2^2 \leq 2. \tag{14}$$

Conditioning on this event, we can rewrite eqn. (12) as follows:

$$\|\mathbf{X}\mathbf{S}\tilde{\mathbf{x}}\|_2^2 \leq 2\|\mathbf{X}\mathbf{S}\mathbf{v}\|_2^2 = 2\left(\mathbf{S}\mathbf{v}\right)^\top \mathbf{X}^\top \mathbf{X}(\mathbf{S}\mathbf{v}) = 2\,\mathbf{y}^\top \mathbf{X}^\top \mathbf{X}\mathbf{y}. \tag{15}$$

Combining eqns. (11) and (15), we conclude

$$\mathbf{y}^\top \mathbf{X}^\top \mathbf{X} \mathbf{y} \geq \frac{1}{2} \mathcal{Z}^* - \frac{\epsilon}{\sqrt{k}} \sqrt{\mathcal{Z}^*} \cdot \|\mathbf{X}\|_F .$$

Using $\mathbf{X}^\top \mathbf{X} = \mathbf{A}$ and $\mathrm{Tr}(\mathbf{A}) = \|\mathbf{X}\|_F^2$ concludes the proof of eqn. (4). Finally, following the lines of eqn. (13), we can prove

$$\mathbb{E}(\|\mathbf{y}\|_2^2) = \mathbb{E}(\|\mathbf{S}\mathbf{v}\|_2^2) = 1.$$

To conclude the proof of the theorem, notice that the failure probability is at most $1/4 + 1/2 = 3/4$ from a union bound on the failure probabilities of eqns. (9) and (14). $\qquad\square$

## A.2 SPCA VIA THRESHOLDING: PROOFS

We will use the notation of Section 3. For notational convenience, let $\sigma_1, \ldots, \sigma_n$ be the diagonal entries of the matrix $\boldsymbol{\Sigma} \in \mathbb{R}^{n \times n}$, i.e., the singular values of $\mathbf{A}$.

**Proof of Lemma 3.2:** Let $\mathbf{U}_{\ell,\perp} \in \mathbb{R}^{n \times (n-\ell)}$ be a matrix whose columns form a basis for the subspace perpendicular to the subspace spanned by the columns of $\mathbf{U}_\ell$. Similarly, let $\boldsymbol{\Sigma}_{\ell,\perp} \in \mathbb{R}^{(n-\ell) \times (n-\ell)}$ be the diagonal matrix of the bottom $n - \ell$ singular values of $\mathbf{A}$. Notice that $\mathbf{U} = [\mathbf{U}_\ell \ \ \mathbf{U}_{\ell,\perp}]$ and $\boldsymbol{\Sigma} = [\boldsymbol{\Sigma}_\ell \ \mathbf{0}; \ \mathbf{0} \ \ \boldsymbol{\Sigma}_{\ell,\perp}]$; thus,

$$\mathbf{U} \boldsymbol{\Sigma}^{1/2} \mathbf{U}^\top = \mathbf{U}_\ell \boldsymbol{\Sigma}_\ell^{1/2} \mathbf{U}_\ell^\top + \mathbf{U}_{\ell,\perp} \boldsymbol{\Sigma}_{\ell,\perp}^{1/2} \mathbf{U}_{\ell,\perp}^\top.$$

By the Pythagorean theorem,

$$\left\| \mathbf{U} \boldsymbol{\Sigma}^{1/2} \mathbf{U}^\top \mathbf{x} \right\|_2^2 = \left\| \mathbf{U}_\ell \boldsymbol{\Sigma}_\ell^{1/2} \mathbf{U}_\ell^\top \mathbf{x} \right\|_2^2 + \left\| \mathbf{U}_{\ell,\perp} \boldsymbol{\Sigma}_{\ell,\perp}^{1/2} \mathbf{U}_{\ell,\perp}^\top \mathbf{x} \right\|_2^2 .$$

Using invariance properties of the vector two-norm and sub-multiplicativity, we get

$$\left\| \boldsymbol{\Sigma}_\ell^{1/2} \mathbf{U}_\ell^\top \mathbf{x} \right\|_2^2 \geq \left\| \boldsymbol{\Sigma}^{1/2} \mathbf{U}^\top \mathbf{x} \right\|_2^2 - \left\| \boldsymbol{\Sigma}_{\ell,\perp}^{1/2} \right\|_2^2 \left\| \mathbf{U}_{\ell,\perp}^\top \mathbf{x} \right\|_2^2 .$$

We conclude the proof by noting that $\left\| \boldsymbol{\Sigma}^{1/2} \mathbf{U}^\top \mathbf{x} \right\|_2^2 = \mathbf{x}^\top \mathbf{U} \boldsymbol{\Sigma} \mathbf{U}^\top \mathbf{x} = \mathbf{x}^\top \mathbf{A} \mathbf{x}$ and

$$\left\| \boldsymbol{\Sigma}_{\ell,\perp}^{1/2} \right\|_2^2 = \sigma_{\ell+1} \leq \frac{1}{\ell} \sum_{i=1}^n \sigma_i = \frac{\mathrm{Tr}(\mathbf{A})}{\ell} .$$

The inequality above follows since $\sigma_1 \geq \sigma_2 \geq \ldots \sigma_\ell \geq \sigma_{\ell+1} \geq \ldots \geq \sigma_n$. We conclude the proof by setting $\ell = 1/\epsilon$. $\qquad\blacksquare$

**Proof of Theorem 3.1.** Let $R = \{i_1, \ldots, i_{|R|}\}$ be the set of indices of rows of $\mathbf{U}_\ell$ (columns of $\mathbf{U}_\ell^\top$) that have squared norms at least $\epsilon/k$ and let $\bar{R}$ be its complement. Here $|R|$ denotes the cardinality of the set $R$ and $R \cup \bar{R} = \{1, \ldots, n\}$. Let $\mathbf{R} \in \mathbb{R}^{n \times |R|}$ be the sampling matrix that selects the columns of $\mathbf{U}_\ell$ whose indices are in the set $R$ and let $\mathbf{R}_\perp \in \mathbb{R}^{n \times (n-|R|)}$ be the sampling matrix that selects the columns of $\mathbf{U}_\ell$ whose indices are in the set $\bar{R}$. Thus, each column of $\mathbf{R}$ (respectively $\mathbf{R}_\perp$) has a single non-zero entry, equal to one, corresponding to one of the $|R|$ (respectively $|\bar{R}|$) selected columns. Formally, $\mathbf{R}_{i_t,t} = 1$ for all $t = 1, \ldots, |R|$, while all other entries of $\mathbf{R}$ (respectively $\mathbf{R}_\perp$) are set to zero; $\mathbf{R}_\perp$ can be defined analogously. The following properties are easy to prove: $\mathbf{R}\mathbf{R}^\top + \mathbf{R}_\perp \mathbf{R}_\perp^\top = \mathbf{I}_n$; $\mathbf{R}^\top \mathbf{R} = \mathbf{I}$; $\mathbf{R}_\perp^\top \mathbf{R}_\perp = \mathbf{I}$; $\mathbf{R}_\perp^\top \mathbf{R} = \mathbf{0}$. Recall that $\mathbf{x}^*$ is the optimal solution to the SPCA problem from eqn. (1). We proceed as follows:

$$\begin{aligned} \left\| \boldsymbol{\Sigma}_\ell^{1/2} \mathbf{U}_\ell^\top \mathbf{x} \right\|_2^2 &= \left\| \boldsymbol{\Sigma}_\ell^{1/2} \mathbf{U}_\ell^\top (\mathbf{R}\mathbf{R}^\top + \mathbf{R}_\perp \mathbf{R}_\perp^\top) \mathbf{x} \right\|_2^2 \\ &\leq 2 \left\| \boldsymbol{\Sigma}_\ell^{1/2} \mathbf{U}_\ell^\top \mathbf{R} \mathbf{R}^\top \mathbf{x}^* \right\|_2^2 + 2 \left\| \boldsymbol{\Sigma}_\ell^{1/2} \mathbf{U}_\ell^\top \mathbf{R}_\perp \mathbf{R}_\perp^\top \mathbf{x}^* \right\|_2^2 \\ &\leq 2 \left\| \boldsymbol{\Sigma}_\ell^{1/2} \mathbf{U}_\ell^\top \mathbf{R} \mathbf{R}^\top \mathbf{x}^* \right\|_2^2 + 2\sigma_1 \left\| \mathbf{U}_\ell^\top \mathbf{R}_\perp \mathbf{R}_\perp^\top \mathbf{x}^* \right\|_2^2 . \end{aligned} \qquad (16)$$

The above inequalities follow from the Pythagorean theorem and sub-multiplicativity. We now bound the second term in the right-hand side of the above inequality.

$$\left\| \mathbf{U}_\ell^\top \mathbf{R}_\perp \mathbf{R}_\perp^\top \mathbf{x}^* \right\|_2 = \| \sum_{i=1}^n (\mathbf{U}_\ell^\top \mathbf{R}_\perp)_{*i} (\mathbf{R}_\perp^\top \mathbf{x}^*)_i \|_2$$

$$\leq \sum_{i=1}^{n} \|(\mathbf{U}_\ell^\top \mathbf{R}_\perp)_{*i}\|_2 \cdot |(\mathbf{R}_\perp^\top \mathbf{x}^*)_i| \leq \sqrt{\frac{\epsilon}{k}} \sum_{i=1}^{n} |(\mathbf{R}_\perp^\top \mathbf{x}^*)_i|$$

$$\leq \sqrt{\frac{\epsilon}{k}} \|\mathbf{R}_\perp^\top \mathbf{x}^*\|_1 \leq \sqrt{\frac{\epsilon}{k}} \sqrt{k} = \sqrt{\epsilon}. \tag{17}$$

In the above derivations we use standard properties of norms and the fact that the columns of $\mathbf{U}_\ell^\top$ that have indices in the set $\bar{R}$ have squared norm at most $\epsilon/k$. The last inequality follows from $\|\mathbf{R}_\perp^\top \mathbf{x}^*\|_1 \leq \|\mathbf{x}^*\|_1 \leq \sqrt{k}$, since $\mathbf{x}^*$ has at most $k$ non-zero entries and Euclidean norm at most one.

Recall that the vector $\mathbf{y}$ of Algorithm 3 maximizes $\|\mathbf{\Sigma}_\ell^{1/2} \mathbf{U}_\ell^\top \mathbf{R} \mathbf{x}\|_2$ over all vectors $\mathbf{x}$ of appropriate dimensions (including $\mathbf{R}\mathbf{x}^*$) and thus

$$\|\mathbf{\Sigma}_\ell^{1/2} \mathbf{U}_\ell^\top \mathbf{R} \mathbf{y}\|_2 \geq \left\| \mathbf{\Sigma}_\ell^{1/2} \mathbf{U}_\ell^\top \mathbf{R} \mathbf{R}^\top \mathbf{x}^* \right\|_2. \tag{18}$$

Combining eqns. (16), (17), and (18), we get

$$\frac{1}{2} \left\| \mathbf{\Sigma}_\ell^{1/2} \mathbf{U}_\ell^\top \mathbf{x}^* \right\|_2^2 \leq \|\mathbf{\Sigma}_\ell^{1/2} \mathbf{U}_\ell^\top \mathbf{z}\|_2^2 + \epsilon \operatorname{Tr}(\mathbf{A}). \tag{19}$$

In the above we used $\mathbf{z} = \mathbf{R}\mathbf{y}$ (as in Algorithm 3) and $\sigma_1 \leq \operatorname{Tr}(\mathbf{A})$. Notice that

$$\mathbf{U}_\ell \mathbf{\Sigma}_\ell^{1/2} \mathbf{U}_\ell^\top \mathbf{z} + \mathbf{U}_{\ell,\perp} \mathbf{\Sigma}_{\ell,\perp}^{1/2} \mathbf{U}_{\ell,\perp}^\top \mathbf{z} = \mathbf{U} \mathbf{\Sigma}^{1/2} \mathbf{U}^\top \mathbf{z},$$

and use the Pythagorean theorem to get

$$\|\mathbf{U}_\ell \mathbf{\Sigma}_\ell^{1/2} \mathbf{U}_\ell^\top \mathbf{z}\|_2^2 + \|\mathbf{U}_{\ell,\perp} \mathbf{\Sigma}_{\ell,\perp}^{1/2} \mathbf{U}_{\ell,\perp}^\top \mathbf{z}|_2^2 = \|\mathbf{U} \mathbf{\Sigma}^{1/2} \mathbf{U}^\top \mathbf{z}\|_2^2.$$

Using the unitary invariance of the two norm and dropping a non-negative term, we get the bound

$$\|\mathbf{\Sigma}_\ell^{1/2} \mathbf{U}_\ell^\top \mathbf{z}\|_2^2 \leq \|\mathbf{\Sigma}^{1/2} \mathbf{U}^\top \mathbf{z}\|_2^2. \tag{20}$$

Combining eqns. (20) and (19), we conclude

$$\frac{1}{2} \left\| \mathbf{\Sigma}_\ell^{1/2} \mathbf{U}_\ell^\top \mathbf{x}^* \right\|_2^2 \leq \|\mathbf{\Sigma}^{1/2} \mathbf{U}^\top \mathbf{z}\|_2^2 + \epsilon \operatorname{Tr}(\mathbf{A}). \tag{21}$$

We now apply Lemma 3.2 to the optimal vector $\mathbf{x}^*$ to get

$$\left\| \mathbf{\Sigma}^{1/2} \mathbf{U}^\top \mathbf{x}^* \right\|_2^2 - \epsilon \operatorname{Tr}(\mathbf{A}) \leq \left\| \mathbf{\Sigma}_\ell^{1/2} \mathbf{U}_\ell^\top \mathbf{x}^* \right\|_2^2.$$

Combining with eqn. (21) we get

$$\mathbf{z}^\top \mathbf{A} \mathbf{z} \geq \frac{1}{2} \mathcal{Z}^* - \frac{3}{2} \epsilon \operatorname{Tr}(\mathbf{A}).$$

In the above we used $\|\mathbf{\Sigma}^{1/2} \mathbf{U}^\top \mathbf{z}\|_2^2 = \mathbf{z}^\top \mathbf{A} \mathbf{z}$ and $\left\| \mathbf{\Sigma}^{1/2} \mathbf{U}^\top \mathbf{x}^* \right\|_2^2 = (\mathbf{x}^*)^\top \mathbf{A} \mathbf{x}^* = \mathcal{Z}^*$. ∎

### A.3 SPCA VIA A SEMIDEFINITE PROGRAMMING RELAXATION: PROOFS

We start with a lemma arguing that the sparsification procedure of Algorithm 5 does not significantly distort the $\ell_2$ norm of the input/output vectors.

**Lemma A.3** *Let $\mathbf{y}$ and $\mathbf{z}$ be defined as in Algorithm 5. If $\|\mathbf{y}\|_1 \leq \alpha$, then, with probability at least $15/16$,*

$$\|\mathbf{z} - \mathbf{y}\|_2^2 \leq \frac{16\alpha^2}{s}.$$

**Proof :** Notice that

$$\mathbb{E}\left[\|\mathbf{z} - \mathbf{y}\|_2^2\right] = \sum_{i=1}^{n} \left(\frac{1}{p_i} - 1\right) y_i^2 \leq \sum_{i=1}^{n} \frac{y_i^2}{p_i} \leq \|\mathbf{y}\|_1 \sum_{i=1}^{n} \frac{y_i}{s} = \frac{\|\mathbf{y}\|_1^2}{s},$$

which is at most $\frac{\alpha^2}{s}$ from our assumptions. The lemma follows by Markov's inequality. □

To prove Lemma 4.3, we start with the following consequence of the triangle inequality, e.g., Lemma 3 from (Fountoulakis et al., 2017).

**Lemma A.4** *Let* $\mathbf{y}$ *and* $\mathbf{z}$ *be defined as in* Algorithm 5. *Then*

$$|\mathbf{y}^\top \mathbf{A}\mathbf{y} - \mathbf{z}^\top \mathbf{A}\mathbf{z}| \leq 2|\mathbf{y}^\top \mathbf{A}(\mathbf{y}-\mathbf{z})| + |(\mathbf{y}-\mathbf{z})^\top \mathbf{A}(\mathbf{y}-\mathbf{z})|.$$

**Proof :**  This is Lemma 3 from (Fountoulakis et al., 2017). $\qquad\square$

We now proceed to upper bound the two terms in the above lemma separately. The following lemma bounds the first term.

**Lemma A.5** *Let* $\mathbf{y}$ *and* $\mathbf{z}$ *be defined as in* Algorithm 5. *If* $\|\mathbf{y}\|_1 \leq \alpha$ *and* $\|\mathbf{y}\|_2 \leq \beta$, *then, with probability at least* $^{15}/_{16}$, $|\mathbf{y}^\top \mathbf{A}(\mathbf{y}-\mathbf{z})| \leq {4\alpha\beta}/{\sqrt{s}}$.

**Proof :**  Recall that we set $z_i = y_i/p_i$ with probability $p_i$ and zero otherwise, for all $i = 1 \dots n$. Then,

$$\mathbb{E}\left[(\mathbf{y}^\top \mathbf{A}(\mathbf{y}-\mathbf{z}))^2\right] = \sum_{i=1}^{n}\left(\frac{1}{p_i}-1\right)y_i^2(\mathbf{A}_{i*}\mathbf{y})^2.$$

Using $|\mathbf{A}_{i*}\mathbf{y}| \leq \|\mathbf{A}_{i*}\|_2 \|\mathbf{y}\|_2 \leq \beta$ (from our assumption on the $\ell_2$ norm of $\mathbf{y}$ as well as our assumption that the rows/columns of $\mathbf{A}$ have unit norm), it follows that

$$\mathbb{E}\left[(\mathbf{y}^\top \mathbf{A}(\mathbf{y}-\mathbf{z}))^2\right] \leq \beta^2 \sum_{i=1}^{n}\frac{y_i^2}{p_i} \leq \frac{\beta^2\|\mathbf{y}\|_1^2}{s} \leq \frac{\alpha^2\beta^2}{s}.$$

The lemma follows from Markov's inequality. $\qquad\square$

The next lemma provides an upper bound for the second term in the right hand side of Lemma A.4.

**Lemma A.6** *Let* $\mathbf{y}$ *and* $\mathbf{z}$ *be defined as in* Algorithm 5. *If* $\|\mathbf{y}\|_1 \leq \alpha$ *and* $\|\mathbf{y}\|_2 \leq \beta$, *then, with probability at least* $^{15}/_{16}$,

$$|(\mathbf{y}-\mathbf{z})^\top \mathbf{A}(\mathbf{y}-\mathbf{z})| \leq \sqrt{\frac{64\alpha^4}{s^2} + \frac{96\alpha\beta^3}{s}}.$$

**Proof :**  Let $\zeta_i = \frac{1}{p_i}$ with probability $p_i$ and zero otherwise for all $i = 1 \dots n$. Then,

$$\mathbb{E}\left[\left((\mathbf{y}-\mathbf{z})^\top \mathbf{A}(\mathbf{z}-\mathbf{y})\right)^2\right] = \sum_{a,b,c,d} A_{a,c}A_{b,d}y_ay_by_cy_d \cdot \mathbb{E}\left[(1-\zeta_a)(1-\zeta_b)(1-\zeta_c)(1-\zeta_d)\right].$$

We immediately have $\mathbb{E}\left[1-\zeta_i\right] = 0$. Thus, if any of the indices $a, b, c, d$ appears only once in the above summation, then

$$\mathbb{E}\left[(1-\zeta_a)(1-\zeta_b)(1-\zeta_c)(1-\zeta_d)\right] = 0.$$

Let

$$B_1 = \sum_{a\neq b}A_{a,b}^2 y_a^2 y_b^2 \mathbb{E}\left[(1-\zeta_a)^2(1-\zeta_b)^2\right],$$

$$B_2 = \sum_{a\neq b}A_{a,a}A_{b,b}y_a^2 y_b^2 \mathbb{E}\left[(1-\zeta_a)^2(1-\zeta_b)^2\right],$$

$$B_3 = \sum_{a\neq b}A_{a,b}A_{b,a}y_a^2 y_b^2 \mathbb{E}\left[(1-\zeta_a)^2(1-\zeta_b)^2\right],$$

$$B_4 = \sum_{a=1}^{n}A_{a,a}^2 y_a^4 \mathbb{E}\left[(1-\zeta_a)^4\right].$$

It now follows that

$$\mathbb{E}\left[\left((\mathbf{y}-\mathbf{z})^\top \mathbf{A}(\mathbf{z}-\mathbf{y})\right)^2\right] = \sum_{i=1}^{4} B_i.$$

Using $|A_{i,j}| \leq 1$ for all $i, j$, we can bound $B_1$, $B_2$, and $B_3$ by

$$\max_{i=1,2,3}\{B_i\} \leq \sum_{a=1}^{n}y_a^2 \mathbb{E}\left[(1-\zeta_a)^2\right] \sum_{b=1}^{n}y_b^2 \mathbb{E}\left[(1-\zeta_b)^2\right].$$

Using $\mathbb{E}\left[(1-\zeta_i)^2\right] = \frac{1}{p_i} - 1$ for all $i$, we get

$$\max_{i=1,2,3}\{B_i\}) \leq \left(\sum_{a=1}\left(\frac{1}{p_a}-1\right)y_a^2\right)^2 \leq \left(\frac{\|\mathbf{y}\|_1^2}{s}\right)^2 \leq \frac{\alpha^2}{s^2},$$

where the inequality follows by $\|\mathbf{y}\|_1 \leq \alpha$. To bound $B_4$, use $\|\mathbf{y}\|_1 \leq \alpha$ and $\|\mathbf{y}\|_2 \leq \beta$, to get

$$B_4 = \sum_{a=1}^n A_{a,a}^2 y_a^4 \mathbb{E}\left[(1-\zeta_a)^4\right] \leq \sum_{a=1}^n y_a^4 \mathbb{E}\left[(1-\zeta_a)^4\right] = \sum_{a=1}^n y_a^4\left(\frac{1}{p_a^3} - \frac{4}{p_a^2} + \frac{6}{p_a} - 4 + p_a\right)$$

$$\leq \sum_{a=1}^n y_a^4\left(\frac{\|\mathbf{y}\|_1^3}{|y_a^3|s^3} + \frac{6\|\mathbf{y}\|_1}{|y_a|s}\right) \leq \frac{\|\mathbf{y}\|_1^4}{s^3} + \frac{6\|\mathbf{y}\|_3\|\mathbf{y}\|_2^3}{s} \leq \frac{\alpha^4}{s^3} + \frac{6\alpha\beta^3}{s}.$$

The last inequality follows from properties of norms, namely that $\|\mathbf{y}\|_3^3 \leq \|\mathbf{y}\|_2^3$. Thus,

$$\mathbb{E}\left[\left((\mathbf{y}-\mathbf{z})^\top\mathbf{A}(\mathbf{z}-\mathbf{y})\right)^2\right] \leq \frac{\alpha^4}{s^3} + \frac{3\alpha^4}{s^2} + \frac{6\alpha\beta^3}{s} \leq \frac{4\alpha^4}{s^2} + \frac{6\alpha\beta^3}{s}.$$

Using Markov's inequality, we conclude

$$\mathbf{Pr}\left[|(\mathbf{z}-\mathbf{y})^\top\mathbf{A}(\mathbf{z}-\mathbf{y})| \geq 4\sqrt{\frac{4\alpha^4}{s^2} + \frac{6\alpha\beta^3}{s}}\right] \leq \frac{1}{16}.$$

$\square$

**Proof of Lemma 4.2:**  Observe that Lemma 4.2 follows immediately from Lemma A.4, Lemma A.5, Lemma A.6, and a union bound. $\square$

The next two lemmas bound the $\ell_2$ and $\ell_1$ norms of the vectors $\mathbf{y}_i$ for all $i = 1\ldots M$. We will bound the norm of a single vector $\mathbf{y}_i$ (we will drop the index) and then apply a union bound on all $M$ vectors.

**Lemma A.7** *Let $\mathbf{y}$ be defined as in Algorithm 5. Then,*

$$\mathbf{Pr}\left[\|\mathbf{y}\|_2 \geq 2\sqrt{\log M}\right] \leq \frac{1}{M^2}.$$

**Proof :**  Let $\mathbf{Z} = \mathbf{U}\boldsymbol{\Sigma}\mathbf{V}^\top$ be the singular value decomposition of $\mathbf{Z}$ and let $\sigma_i = \boldsymbol{\Sigma}_{i,i}$, for $i = 1\ldots n$, be the singular values of $\mathbf{Z}$. Since $\mathrm{Tr}(\mathbf{Z}) = 1$, it follows that $\sum_{i=1}^n \sigma_i = 1$ and also $\sigma_i \leq 1$ for all $i = 1\ldots n$. Additionally,

$$\sum_{i=1}^n \sigma_i^2 \leq \sum_{i=1}^n \sigma_i \leq 1. \tag{22}$$

Then,

$$\|\mathbf{y}\|_2^2 = \|\mathbf{Z}\mathbf{g}\|_2^2 = \mathbf{g}^\top\mathbf{Z}^\top\mathbf{Z}\mathbf{g} = \mathbf{g}^\top\mathbf{V}\boldsymbol{\Sigma}^2\mathbf{V}^\top\mathbf{g} = \left\|\boldsymbol{\Sigma}\mathbf{V}^\top\mathbf{g}\right\|_2^2.$$

The rotational invariance of the Gaussian distribution implies that $\mathbf{y}_2 \sim \mathbf{h}_2$, where $\mathbf{h}$ is a random vector whose $i$-th entry $h_i$ satisfies $h_i \sim \mathcal{N}(0,\sigma_i^2)$. Hence,

$$\mathbb{E}\left[\|\mathbf{y}\|_2^2\right] = \mathbb{E}\left[\|\mathbf{h}\|_2^2\right] = \sum_{i=1}^n \sigma_i^2 \leq 1.$$

Now, from Markov's inequality, for any $C > 0$,

$$\mathbf{Pr}\left[\|\mathbf{y}\|_2 \geq t + \frac{\log M}{t}\right] = \mathbf{Pr}\left[e^{C\|\mathbf{y}\|_2} \geq e^{Ct + C\log M/t}\right] \leq \frac{\mathbb{E}\left[e^{C\|\mathbf{y}\|_2}\right]}{e^{Ct}M^{C/t}}.$$

Then,

$$\mathbb{E}\left[e^{C\|\mathbf{y}\|_2}\right] = \mathbb{E}\left[e^{C\|\mathbf{h}\|_2}\right]$$

$$\leq \prod_{i=1}^{n} \frac{2}{\sqrt{2\pi}\sigma_i} \int_0^{\infty} e^{Cx} e^{-x^2/2\sigma_i^2} \, dx$$

$$= \prod_{i=1}^{n} \left( \frac{2}{\sqrt{2\pi}\sigma_i} e^{C^2\sigma_i^2/2} \int_0^{\infty} e^{-\left(\frac{x}{\sqrt{2}\sigma_i} - \frac{C\sigma_i}{\sqrt{2}}\right)^2} \, dx \right)$$

$$= \prod_{i=1}^{n} \left( \frac{2}{\sqrt{\pi}} e^{C^2\sigma_i^2/2} \int_0^{\infty} e^{-t^2} \, dt \right) = \prod_{i=1}^{n} e^{C^2\sigma_i^2/2} = \exp\left( \sum_{i=1}^{n} C^2\sigma_i^2/2 \right).$$

Using eqn. (22), we get

$$\mathbb{E}\left[ e^{C\|\mathbf{y}\|_2} \right] \leq e^{C^2/2}.$$

Setting $C = 2t$ and $\epsilon \leq 1$, we get

$$\mathbf{Pr}\left[ \|\mathbf{y}\|_2 \geq t + \frac{\log M}{t} \right] \leq \frac{e^{C^2/2}}{e^{Ct} M^{C/t}} = \frac{1}{M^2}.$$

Setting $t = \sqrt{\log M}$ concludes the proof. $\qquad\square$

Prior to bounding the $\ell_1$ norm of $\mathbf{y}$, we present a measure concentration result that will be useful in our proof. First, recall the definition of $L$-Lipschitz functions.

**Definition A.8** *Let $f : \mathbb{R}^n \to \mathbb{R}$ be any function. If $\|f(\mathbf{x}) - f(\mathbf{y})\|_2 \leq L \|\mathbf{x} - \mathbf{y}\|_2$ for all $\mathbf{x}, \mathbf{y} \in \mathbb{R}^n$, then $f$ is $L$-Lipschitz.*

**Theorem A.9 (Gaussian Lipschitz Concentration)** *(Wainwright, 2015) Let $f$ be an $L$-Lipschitz function and let $\mathbf{g} \in \mathbb{R}^n$ be a vector of i.i.d. Gaussians. Then $f(x)$ is sub-Gaussian with variance $L^2$ and, for all $t \geq 0$,*

$$\mathbf{Pr}\left[ |f(x) - \mathbb{E}\left[ f(x) \right]| \geq t \right] \leq 2 e^{-t^2/2L^2}.$$

**Lemma A.10** *Let $\mathbf{y}$ be defined as in Algorithm 5. Then,*

$$\mathbf{Pr}\left[ \|\mathbf{y}\|_1 \geq k(1 + 2k\sqrt{\log M}) \right] \leq \frac{1}{M^2}.$$

**Proof :** Since $g_j \sim \mathcal{N}(0, 1)$ for all $j = 1 \ldots n$, the 2-stability of the Gaussian distribution implies that

$$\mathbb{E}\left[ \|\mathbf{Zg}\|_1 \right] = \sum_{i=1}^{n} \left| \sum_{j=1}^{n} Z_{i,j} g_j \right| = \sum_{i=1}^{n} \sqrt{\frac{2}{\pi}} \|\mathbf{Z}_{i*}\|_2 = \sqrt{\frac{2}{\pi}} \|\mathbf{Z}\|_{1,2}.$$

Let $f(\mathbf{x}) = \|\mathbf{Zx}\|_1$. The triangle inequality implies that

$$\left| \|\mathbf{Zx}\|_1 - \|\mathbf{Zy}\|_1 \right| \leq \sum_{i=1}^{n} |\mathbf{Z}_{i*}\mathbf{x} - \mathbf{Z}_{i*}\mathbf{y}| = \sum_{i=1}^{n} |\mathbf{Z}_{i*}(\mathbf{x} - \mathbf{y})|.$$

Thus, by Cauchy-Schwarz,

$$\left| \|\mathbf{Zx}\|_1 - \|\mathbf{Zy}\|_1 \right| \leq \sum_{i=1}^{n} \|\mathbf{Z}_{i*}\|_2 \|\mathbf{x} - \mathbf{y}\|_2,$$

and $f(\mathbf{x})$ is $\|\mathbf{Z}\|_{1,2}$-Lipschitz[4]. Using Theorem A.9,

$$\mathbf{Pr}\left[ \left| \|\mathbf{y}\|_1 - \sqrt{\frac{2}{\pi}} \|\mathbf{Z}\|_{1,2} \right| \geq t \right] \leq 2 e^{-t^2/2\|\mathbf{Z}\|_{1,2}^2},$$

---

[4]Recall that the $L_{p,q}$ norm of $\mathbf{A}$ is $\|\mathbf{A}\|_{p,q} = \left( \sum_{i=1}^{n} \left( \sum_{j=1}^{n} |A_{i,j}|^q \right)^{\frac{p}{q}} \right)^{\frac{1}{p}}$, e.g., $\|\mathbf{A}\|_F = \|\mathbf{A}\|_{2,2}$.

for all $t \geq 0$. Setting $t = 2\sqrt{\log M}$ and noting that $\|\mathbf{Z}\|_{1,2} \leq \|\mathbf{Z}\|_{1,1} \leq k$, we get

$$\mathbf{Pr}\left[\|\mathbf{y}\|_1 \geq k(1 + 2\sqrt{\log M})\right] \leq \frac{1}{M^2}.$$

$\square$

**Proof of Lemma 4.3:**  Using Lemma A.7 and Lemma A.10, we conclude that $\|\mathbf{y}\|_1 \leq \alpha$ and $\|\mathbf{y}\|_2 \leq \beta$ both hold with probability at least $1 - \frac{2}{M}$. Using Lemma A.4, we get

$$|\mathbf{y}^\top \mathbf{A} \mathbf{y} - \mathbf{z}^\top \mathbf{A} \mathbf{z}| \leq 2|\mathbf{y}^\top \mathbf{A}(\mathbf{y} - \mathbf{z})| + |(\mathbf{y} - \mathbf{z})^\top \mathbf{A}(\mathbf{y} - \mathbf{z})|.$$

Since $|\mathbf{y}^\top \mathbf{A}(\mathbf{y} - \mathbf{z})| \leq \frac{4\alpha\beta}{\sqrt{s}}$ with probability at least $\frac{15}{16}$ (by Lemma A.5) and

$$|(\mathbf{y} - \mathbf{z})^\top \mathbf{A}(\mathbf{y} - \mathbf{z})| \leq \sqrt{\frac{64\alpha^4}{s^2} + \frac{96\alpha\beta^3}{s}}$$

with probability at least $\frac{15}{16}$ (by Lemma A.6), setting $s = \frac{450\alpha^2\beta^3}{\epsilon^2}$, we get

$$\mathbf{y}^\top \mathbf{A} \mathbf{y} \leq \mathbf{z}^\top \mathbf{A} \mathbf{z} + \epsilon,$$

with probability at least $14/16 - 2/M \geq 3/4$, since $M \geq 16$.  ∎  $\square$

We now prove an inequality that was used in eqn. (5) to compare $\mathbf{y}^\top \mathbf{A} \mathbf{y}$ and $\mathrm{Tr}(\mathbf{A}\mathbf{Z})$.

**Lemma A.11** *Let* $\mathbf{Z}, \mathbf{A} \in \mathbb{R}^{n \times n}$ *be PSD matrices and* $\mathrm{Tr}(\mathbf{A}\mathbf{Z}) \leq \alpha\,\mathrm{Tr}(\mathbf{A}\mathbf{Z}_1)$ *for some* $\alpha \geq 1$, *where* $\mathbf{Z}_1$ *is the best rank-1 approximation of* $\mathbf{Z}$. *Then,*

$$\mathrm{Tr}(\mathbf{Z}\mathbf{A}\mathbf{Z}) \geq \gamma_Z \cdot \mathrm{Tr}(\mathbf{A}\mathbf{Z}).$$

*Here* $\gamma_Z = \left(1 - \left(1 - \frac{1}{\kappa(\mathbf{Z})}\right)\left(1 - \frac{1}{\alpha}\right)\right)\sigma_1(\mathbf{Z})$ *with* $\sigma_1(\mathbf{Z})$ *and* $\kappa(\mathbf{Z})$ *being the top singular value and condition number of* $\mathbf{Z}$ *respectively.*

**Proof :**  For simplicity of exposition, assume that $\mathrm{rank}(\mathbf{Z}) = n$. Let $\mathbf{Z} = \mathbf{U}\boldsymbol{\Sigma}\mathbf{U}$ be the SVD of $\mathbf{Z}$. Suppose $\mathbf{U} = (\mathbf{U}_1 \ \ \mathbf{U}_{1,\perp})$ and $\boldsymbol{\Sigma} = \begin{pmatrix} \boldsymbol{\Sigma}_1 & \mathbf{0} \\ \mathbf{0} & \boldsymbol{\Sigma}_{1,\perp} \end{pmatrix}$ such that we have $\mathbf{Z}_1 = \mathbf{U}_1\boldsymbol{\Sigma}_1\mathbf{U}_1^\top$ and $\mathbf{Z}_{1,\perp} = \mathbf{U}_{1,\perp}\boldsymbol{\Sigma}_{1,\perp}\mathbf{U}_{1,\perp}^\top$. As $\mathbf{Z}_1$ is the best rank-1 approximation of $\mathbf{Z}$, we have $\mathbf{Z}_1\mathbf{Z}_{1,\perp} = \mathbf{Z}_{1,\perp}\mathbf{Z}_1 = \mathbf{0}$. Using this, we rewrite $\mathrm{Tr}(\mathbf{Z}\mathbf{A}\mathbf{Z})$ as the following

$$\begin{aligned}
\mathrm{Tr}(\mathbf{Z}\mathbf{A}\mathbf{Z}) &= \mathrm{Tr}\left((\mathbf{Z}_1 + \mathbf{Z}_{1,\perp})\mathbf{A}(\mathbf{Z}_1 + \mathbf{Z}_{1,\perp})\right) \\
&= \mathrm{Tr}(\mathbf{Z}_1\mathbf{A}\mathbf{Z}_1) + \mathrm{Tr}(\mathbf{Z}_{1,\perp}\mathbf{A}\mathbf{Z}_{1,\perp}) + \mathrm{Tr}(\mathbf{Z}_1\mathbf{A}\mathbf{Z}_{1,\perp}) + \mathrm{Tr}(\mathbf{Z}_{1,\perp}\mathbf{A}\mathbf{Z}_1) \\
&= \mathrm{Tr}(\mathbf{Z}_1\mathbf{A}\mathbf{Z}_1) + \mathrm{Tr}(\mathbf{Z}_{1,\perp}\mathbf{A}\mathbf{Z}_{1,\perp}) + \mathrm{Tr}(\mathbf{A}\mathbf{Z}_{1,\perp}\mathbf{Z}_1) + \mathrm{Tr}(\mathbf{A}\mathbf{Z}_1\mathbf{Z}_{1,\perp}) \\
&= \mathrm{Tr}(\mathbf{Z}_1\mathbf{A}\mathbf{Z}_1) + \mathrm{Tr}(\mathbf{Z}_{1,\perp}\mathbf{A}\mathbf{Z}_{1,\perp}),
\end{aligned} \tag{23}$$

where the third equality follows from the invariance of matrix trace under cyclic permutations and the last step is due to $\mathbf{Z}_1\mathbf{Z}_{1,\perp} = \mathbf{Z}_{1,\perp}\mathbf{Z}_1 = \mathbf{0}$.

Next, we rewrite $\mathrm{Tr}(\mathbf{A}\mathbf{Z})$ as

$$\begin{aligned}
\mathrm{Tr}(\mathbf{A}\mathbf{Z}) &= \mathrm{Tr}(\mathbf{A}\mathbf{Z}_1) + \mathrm{Tr}(\mathbf{A}\mathbf{Z}_{1,\perp}) = \mathrm{Tr}(\mathbf{A}\mathbf{Z}_1\mathbf{Z}_1^\dagger\mathbf{Z}_1) + \mathrm{Tr}(\mathbf{A}\mathbf{Z}_{1,\perp}) \\
&= \mathrm{Tr}(\mathbf{Z}_1^\dagger\mathbf{Z}_1\mathbf{A}\mathbf{Z}_1) + \mathrm{Tr}(\mathbf{A}\mathbf{Z}_{1,\perp}) \\
&\leq \sigma_1(\mathbf{Z}_1^\dagger) \cdot \sigma_1(\mathbf{Z}_1\mathbf{A}\mathbf{Z}_1) + \mathrm{Tr}(\mathbf{A}\mathbf{Z}_{1,\perp})
\end{aligned} \tag{24}$$

where $\mathbf{Z}_1^\dagger$ is the pseudo-inverse of $\mathbf{Z}_1$ and we have used the fact $\mathbf{Z}_1\mathbf{Z}_1^\dagger\mathbf{Z}_1 = \mathbf{Z}_1$ and the last inequality follows from the *von Neumann*'s trace inequality. Now, noting that $\sigma_1(\mathbf{Z}_1^\dagger) = \frac{1}{\sigma_1(\mathbf{Z})}$ along with the fact that $\sigma_1(\mathbf{Z}_1\mathbf{A}\mathbf{Z}_1) \leq \mathrm{Tr}(\mathbf{Z}_1\mathbf{A}\mathbf{Z}_1)$ applying eqn. (23), we have

$$\mathrm{Tr}(\mathbf{A}\mathbf{Z}) \leq \frac{1}{\sigma_1(\mathbf{Z})}\left(\mathrm{Tr}(\mathbf{Z}\mathbf{A}\mathbf{Z}) - \mathrm{Tr}(\mathbf{Z}_{1,\perp}\mathbf{A}\mathbf{Z}_{1,\perp})\right) + \mathrm{Tr}(\mathbf{A}\mathbf{Z}_{1,\perp}) \tag{25}$$

Next, we will show that $\text{Tr}(\mathbf{Z}_{1,\perp}\mathbf{A}\mathbf{Z}_{1,\perp}) \geq \sigma_n(\mathbf{Z})\cdot\text{Tr}(\mathbf{A}\mathbf{Z}_{1,\perp})$. First, note that $\boldsymbol{\Sigma}_{1,\perp} \succcurlyeq \sigma_n(\mathbf{Z})\cdot\mathbf{I}_{n-1}$, as $\sigma_n(\mathbf{Z}) \leq \sigma_i(\mathbf{Z})$ for all $i = 2, \ldots, n$. Therefore, pre- and post-multiplying both sides by $\boldsymbol{\Sigma}_{1,\perp}^{1/2}$, we further have $\boldsymbol{\Sigma}_{1,\perp}^2 \succcurlyeq \sigma_n(\mathbf{Z}) \cdot \boldsymbol{\Sigma}_{1,\perp}$. Again, pre- and post-multiplying both sides by $\mathbf{U}_{1,\perp}$ and $\mathbf{U}_{1,\perp}^\top$, we have:

$$\mathbf{Z}_{1,\perp}^2 = \mathbf{U}_{1,\perp}\boldsymbol{\Sigma}_{1,\perp}^2\mathbf{U}_{1,\perp}^\top \succcurlyeq \sigma_n(\mathbf{Z}) \cdot \mathbf{U}_{1,\perp}\boldsymbol{\Sigma}_{1,\perp}\mathbf{U}_{1,\perp}^\top = \sigma_n(\mathbf{Z}) \cdot \mathbf{Z}_{1,\perp} \,. \tag{26}$$

As the matrix $\mathbf{A}$ is also PSD, it has a PSD square-root $\mathbf{A}^{1/2}$ such that $\mathbf{A} = \mathbf{A}^{1/2} \cdot \mathbf{A}^{1/2}$. Now, pre- and post-multiplying both sides of eqn. (26) by $\mathbf{A}^{1/2}$, we have

$$\mathbf{A}^{1/2}\mathbf{Z}_{1,\perp}^2\mathbf{A}^{1/2} \succcurlyeq \sigma_n(\mathbf{Z}) \cdot \mathbf{A}^{1/2}\mathbf{Z}_{1,\perp}\mathbf{A}^{1/2} \tag{27}$$

Next, we rewrite $\text{Tr}(\mathbf{Z}_{1,\perp}\mathbf{A}\mathbf{Z}_{1,\perp})$ as follows

$$\text{Tr}(\mathbf{Z}_{1,\perp}\mathbf{A}\mathbf{Z}_{1,\perp}) = \text{Tr}(\mathbf{A}\mathbf{Z}_{1,\perp}^2) = \text{Tr}(\mathbf{A}^{1/2}\mathbf{Z}_{1,\perp}^2\mathbf{A}^{1/2}) = \sum_{i=1}^n \mathbf{e}_i^\top\mathbf{A}^{1/2}\mathbf{Z}_{1,\perp}^2\mathbf{A}^{1/2}\mathbf{e}_i$$

$$\geq \sigma_n(\mathbf{Z}) \cdot \sum_{i=1}^n \mathbf{e}_i^\top\mathbf{A}^{1/2}\mathbf{Z}_{1,\perp}\mathbf{A}^{1/2}\mathbf{e}_i = \sigma_n(\mathbf{Z}) \cdot \text{Tr}\left(\mathbf{A}^{1/2}\mathbf{Z}_{1,\perp}\mathbf{A}^{1/2}\right)$$

$$= \sigma_n(\mathbf{Z}) \cdot \text{Tr}\left(\mathbf{A}\mathbf{Z}_{1,\perp}\right) \,. \tag{28}$$

In the above, $\mathbf{e}_1, \mathbf{e}_2, \ldots, \mathbf{e}_n \in \mathbf{R}n$ are the canonical basis vectors and we have used the invariance property of matrix trace under cyclic permutations. Finally, the inequality in eqn. (28) directly follows from eqn. (27), as eqn. (27) boils down to $\mathbf{x}^\top\mathbf{A}^{1/2}\mathbf{Z}_{1,\perp}^2\mathbf{A}^{1/2}\mathbf{x} \geq \sigma_n(\mathbf{Z}) \cdot \mathbf{x}^\top\mathbf{A}^{1/2}\mathbf{Z}_{1,\perp}\mathbf{A}^{1/2}\mathbf{x}$ for any vector $\mathbf{x} \neq \mathbf{0}$.

Next, we combine eqns. (25) and (28) and replacing $\kappa(\mathbf{Z}) = \frac{\sigma_1(\mathbf{Z})}{\sigma_n(\mathbf{Z})}$ to get

$$\text{Tr}(\mathbf{A}\mathbf{Z}) \leq \frac{\text{Tr}(\mathbf{Z}\mathbf{A}\mathbf{Z})}{\sigma_1(\mathbf{Z})} + \left(1 - \frac{1}{\kappa(\mathbf{Z})}\right)\text{Tr}(\mathbf{A}\mathbf{Z}_{1,\perp})$$

$$= \frac{\text{Tr}(\mathbf{Z}\mathbf{A}\mathbf{Z})}{\sigma_1(\mathbf{Z})} + \left(1 - \frac{1}{\kappa(\mathbf{Z})}\right)(\text{Tr}(\mathbf{A}\mathbf{Z}) - \text{Tr}(\mathbf{A}\mathbf{Z}_1))$$

$$\leq \frac{\text{Tr}(\mathbf{Z}\mathbf{A}\mathbf{Z})}{\sigma_1(\mathbf{Z})} + \left(1 - \frac{1}{\kappa(\mathbf{Z})}\right)\left(1 - \frac{1}{\alpha}\right)\text{Tr}(\mathbf{A}\mathbf{Z})\,, \tag{29}$$

where the equality is holds as $\text{Tr}(\mathbf{A}\mathbf{Z}) = \text{Tr}(\mathbf{A}\mathbf{Z}_1) + \text{Tr}(\mathbf{A}\mathbf{Z}_{1,\perp})$ and the last inequality is due to our assumption that $\text{Tr}(\mathbf{A}\mathbf{Z}) \leq \alpha\,\text{Tr}(\mathbf{A}\mathbf{Z}_1)$. This concludes the proof. □

To finalize our proof, we use the following result of (Avron & Toledo, 2011) for estimating the trace of PSD matrices.

**Theorem A.12** *(Avron & Toledo, 2011) Given a PSD matrix $\mathbf{A} \in \mathbb{R}^{n\times n}$, let $M = 80/\epsilon^2$. Let $\mathbf{g}_i$ (for $i = 1 \ldots M$) be standard Gaussian random vectors. Then with probability at least $7/8$,*

$$\left|\text{Tr}(\mathbf{A}) - \frac{1}{M}\sum_{i=1}^M \mathbf{g}_i^\top\mathbf{A}\mathbf{g}_i\right| \leq \epsilon \cdot \text{Tr}(\mathbf{A}).$$

We are now ready to prove the correctness of Algorithm 5 by establishing Theorem A.13.

**Theorem A.13** *Let $\mathbf{Z}$ be an optimal solution to the relaxed SPCA problem of eqn. (2) and Assume that $\text{Tr}(\mathbf{A}\mathbf{Z}) \leq \alpha\,\text{Tr}(\mathbf{A}\mathbf{Z}_1)$ for some constant $\alpha \geq 1$, where $\mathbf{Z}_1$ is the best rank-1 approximation of $\mathbf{Z}$. Then, there exists an algorithm that takes as input a PSD matrix $\mathbf{A} \in \mathbb{R}^{n\times n}$, an approximation parameter $\epsilon > 0$, and a parameter $k$, and outputs a vector $\mathbf{z}$ such that with probability at least $5/8$,*

$$\mathbb{E}\left[\|\mathbf{z}\|_0\right] \leq s = \frac{450\alpha^2\beta^3}{\epsilon^2}, \qquad \|\mathbf{z}\|_2 \leq \beta + \frac{\alpha}{\sqrt{s}}, \qquad \mathbf{z}^\top\mathbf{A}\mathbf{z} \geq \gamma_Z(1 - \epsilon) \cdot \mathcal{Z}^* - \epsilon.$$

*In the above, $\alpha = k(1 + 2\sqrt{\log M})$, $\beta = 2\sqrt{\log M}$, and $M = 80/\epsilon^2$, and $\gamma_Z = \left(1 - \left(1 - \frac{1}{\kappa(\mathbf{Z})}\right)\left(1 - \frac{1}{\alpha}\right)\right)\sigma_1(\mathbf{Z})$ with $\sigma_1(\mathbf{Z})$ and $\kappa(\mathbf{Z})$ being the top singular value and condition number of $\mathbf{Z}$ respectively.*

**Proof :** Consider Algorithm 5 and let $\mathbf{Z}^*$ be an optimal solution to the SPCA semidefinite relaxation of eqn. (2). Then, as already discussed, $(\mathbf{x}^*)^\top \mathbf{A} \mathbf{x}^* \leq \text{Tr}(\mathbf{A}\mathbf{Z}^*)$, where $\mathbf{x}^*$ is the optimal solution to the SPCA problem of eqn. (1). Then, using Lemma A.11, it follows that

$$\gamma_{Z^*} \text{Tr}(\mathbf{A}\mathbf{Z}^*) \leq \text{Tr}\left((\mathbf{Z}^*)^\top \mathbf{A} \mathbf{Z}^*\right).$$

Applying Theorem A.12 to the matrix $(\mathbf{Z}^*)^\top \mathbf{A} \mathbf{Z}^*$ and using our choice of $\mathbf{y}$ in Algorithm 4, we get

$$\mathbf{y}^\top \mathbf{A} \mathbf{y} \geq \frac{1}{M} \sum_{i=1}^M \mathbf{g}_i^\top (\mathbf{Z}^*)^\top \mathbf{A} \mathbf{Z}^* \mathbf{g}_i \geq (1 - \epsilon) \text{Tr}\left((\mathbf{Z}^*)^\top \mathbf{A} \mathbf{Z}^*\right),$$

with probability at least $7/8$. By Lemma 4.3, we have $\mathbf{y}^\top \mathbf{A} \mathbf{y} \leq \mathbf{z}^\top \mathbf{A} \mathbf{z} + \epsilon$ with probability at least $3/4$. Thus, with probability at least $5/8$,

$$(1 - \epsilon)\gamma_{Z^*} \cdot \mathcal{Z}^* = (1 - \epsilon)\gamma_{Z^*} \cdot (\mathbf{x}^*)^\top \mathbf{A} \mathbf{x}^* \leq \mathbf{z}^\top \mathbf{A} \mathbf{z} + \epsilon.$$

To conclude the proof, we need to bound the $\ell_2$ norm of the solution vector $\mathbf{z}$. Let $\mathcal{E}$ be the event that $\|\mathbf{Z}\mathbf{g}_i\|_1 \leq k(1 + 2\sqrt{\log M})$ *and* $\|\mathbf{Z}\mathbf{g}_i\|_2 \leq 2\sqrt{\log M}$ for all $i = 1 \ldots M$. From Lemma A.7 and Lemma A.10 and the union bound, we have $\mathbf{Pr}\left[\mathcal{E}\right] \geq 1 - \frac{2}{M}$. Conditioned on $\mathcal{E}$, Lemma A.3 implies that, with probability at least $15/16$,

$$\|\mathbf{y} - \mathbf{z}\|_2^2 \leq \frac{16k^2(1 + 2\sqrt{\log M})^2}{s}.$$

Therefore, with probability at least $\frac{15}{16} - \frac{2}{M} \geq \frac{3}{4}$ (since $M \geq 16$), an application of the triangle inequality gets

$$\|\mathbf{z}\|_2 \leq \|\mathbf{y}\|_2 + \|\mathbf{z} - \mathbf{y}\|_2 \leq 2\sqrt{\log M} + \frac{4k(1 + 2\sqrt{\log M})}{\sqrt{s}}.$$

Using our chosen values for $\alpha$ and $\beta$ concludes the proof. □

# B ADDITIONAL NOTES ON EXPERIMENTS

In addition, we normalize the outputs of Algorithm 2 and Algorithm 5 by keeping the rows and the columns of $\mathbf{A}$ corresponding to the nonzero elements of the output vectors and then getting the top singular vector of the induced matrix and padding it with zeros. The above two considerations make our comparisons fair in terms of function $f(\mathbf{y})$ (see Section 5 for the definition of $f(\mathbf{y})$). For Algorithm 3, we fix the threshold parameter $\ell$ to 30 for human genetic data, as well as for the text data; we set $\ell = 10$ for the gene expression data. Finally, for Algorithm 5, we fix $M$ (the number of random Gaussian vectors) to 300 and we use Python's `cvxpy` package to solve eqn. (2). All the experiments were implemented on a single-core Intel(R) Xeon(R) Gold 6126 CPU @ 2.60GHz.

## B.1 REAL DATA

**Population genetics data.** We use population genetics data from the Human Genome Diversity Panel (Consortium, 2007) and the HAPMAP (Li et al., 2008). In particular, we use the 22 matrices (one for each chromosome) that encode all autosomal genotypes. Each matrix contains 2,240 rows and a varying number of columns that is equal to the number of single nucleotide polymorphisms (SNPs, well-known biallelic loci of genetic variation across the human genome) in the respective chromosome. The columns of each matrix were mean-centered as a preprocessing step. See Table 4 for summary statistics.

**Gene expression data.** We also use a lung cancer gene expression dataset (GSE10072) from from the NCBI Gene Expression Omnibus database (Landi et al., 2008). This dataset contains 107 samples (58 cases and 49 controls) and 22,215 features. Both the population genetics and the gene expression datasets are interesting in the context of sparse PCA beyond numerical evaluations, since the sparse components can be directly interpreted to identify small sets of SNPs or genes that capture the data variance.

**Text classification data.** We also evaluate our algorithms on a text classification dataset used in (Fountoulakis et al., 2017). This consists of two publicly available standard test collections for ad hoc information retrieval system evaluation: the *Cranfield* collection that contains $1,398$ abstracts of aerodynamics journal articles and the CISI (Centre for Inventions and Scientific Information) data that contains $1,460$ information science abstracts. Finally, using these two collections, a sparse, $2,858 \times 12,427$ *document-term* matrix was created using the Text-to-Matrix Generator (TMG) (Zeimpekis & Gallopoulos, 2006), with the entries representing the weight of each term in the corresponding document. See Table 5 for summary statistics.

## B.2 SYNTHETIC DATA

We also use a synthetic dataset generated using the same mechanism as in (Fountoulakis et al., 2017). Specifically, we construct the $m \times n$ matrix $\mathbf{X}$ such that $\mathbf{X} = \mathbf{U\Sigma V}^\top + \mathbf{E}_\sigma$. Here, $\mathbf{E}_\sigma$ is a noise matrix, containing i.i.d. Gaussian elements with zero mean and we set $\sigma = 10^{-3}$; $\mathbf{U} \in \mathbb{R}^{m \times m}$ is a Hadamard matrix with normalized columns; $\mathbf{\Sigma} = (\tilde{\mathbf{\Sigma}} \ \mathbf{0}) \in \mathbb{R}^{m \times n}$ such that $\tilde{\mathbf{\Sigma}} \in \mathbb{R}^{m \times m}$ is a diagonal matrix with $\tilde{\mathbf{\Sigma}}_{11} = 100$ and $\tilde{\mathbf{\Sigma}}_{ii} = e^{-i}$ for $i = 2, \ldots, m$; $\mathbf{V} \in \mathbb{R}^{n \times n}$ such that $\mathbf{V} = \mathbf{G}_n(\theta)\tilde{\mathbf{V}}$, where $\tilde{\mathbf{V}} \in \mathbb{R}^{n \times n}$ is also a Hadamard matrix with normalized columns and

$$\mathbf{G}_n(\theta) = \mathbf{G}(i_1, i_1 + 1, \theta)\,\mathbf{G}(i_2, i_2 + 1, \theta)\ldots\mathbf{G}(i_{n/4}, i_{n/4} + 1, \theta),$$

is a composition of $\frac{n}{4}$ Givens rotation matrices with $i_k = \frac{n}{2} + 2k - 1$ for $k = 1, 2, \ldots, \frac{n}{4}$. Here $\mathbf{G}(i, j, \theta) \in \mathbb{R}^{n \times n}$ be a Givens rotation matrix, which rotates the plane $i - j$ by an angle $\theta$. For $\theta \approx 0.27\pi$ and $n = 2^{12}$, the matrix $\mathbf{G}_n(\theta)$ rotates the bottom $\frac{n}{2}$ components of the columns of $\tilde{\mathbf{V}}$, making half of them almost zero and the rest half larger. Figure 2 shows the absolute values of the elements of the first column of the matrices $\mathbf{V}$ and $\tilde{\mathbf{V}}$.

## B.3 ADDITIONAL EXPERIMENTS

| | $X_1$ | $X_2$ | $X_3$ | $X_4$ | $X_5$ | $X_6$ | $X_7$ | $X_8$ | $X_9$ | $X_{10}$ | PVE | $\mathcal{Z}^*$ |
|---|---|---|---|---|---|---|---|---|---|---|---|---|
| spca-r (Algorithm 2) | 0 | 0 | 0 | 0 | 0.51 | 0.51 | 0.50 | 0 | 0 | 0.48 | 39.6% | 1164.2 |
| spca-d (Algorithm 3) | 0.50 | 0.50 | 0.50 | 0.50 | 0 | 0 | 0 | 0 | 0 | 0 | 39.5% | 1161.0 |
| spca-sdp (Algorithm 5) | 0 | 0 | 0 | 0 | 0.50 | 0.50 | 0.50 | 0.50 | 0 | 0 | 40.9% | 1201.0 |
| dec (Yuan et al., 2019) | 0 | 0 | 0 | 0 | 0.50 | 0.50 | 0.50 | 0.50 | 0 | 0 | 40.9% | 1201.0 |
| cwpca (Beck & Vaisbourd, 2016) | 0.50 | 0.50 | 0.50 | 0.50 | 0 | 0 | 0 | 0 | 0 | 0 | 39.5% | 1161.0 |
| spca-lowrank (Papailiopoulos et al., 2013) | 0 | 0 | 0 | 0 | 0.50 | 0.50 | 0.50 | 0.50 | 0 | 0 | 40.9% | 1200.9 |
| spca (Zou et al., 2006) | 0 | 0 | 0 | 0 | 0.50 | 0.50 | 0.50 | 0.50 | 0 | 0 | 40.9% | 1201.0 |
| dspca (d'Aspremont et al., 2007) | 0 | 0 | 0 | 0 | 0.50 | 0.50 | 0.50 | 0.50 | 0 | 0 | 40.9% | 1201.0 |

Table 2: Loadings, % variance explained (PVE), and the objective function value for the first principal component of the artificial data.

Additionally, in order to further explore the sparsity patterns of the outputs of our algorithms and how close they are as compared to standard methods, we further apply our methods on a simulation example proposed by (Zou et al., 2006). We describe them below:

**Artificial Data of (Zou et al., 2006).** In this example, three *hidden* factors $V_1$, $V_2$, and $V_3$ are created in the following way:

$$V_1 \sim \mathcal{N}(0, 290), \quad V_2 \sim \mathcal{N}(0, 300),$$

$$V_3 = -0.3\,V_1 + 0.925\,V_2 + \varepsilon, \quad \varepsilon \sim \mathcal{N}(0, 1) \text{ and}$$

$$V_1, \ V_2, \text{ and } \varepsilon \text{ are independent.}$$

Next, we create 10 observable variables $X_1, X_2, \ldots, X_{10}$ in the following way:

$$X_i = V_1 + \varepsilon_{i1}, \ \varepsilon_{i1} \sim \mathcal{N}(0, 1), \quad i = 1, 2, 3, 4,$$
$$X_i = V_2 + \varepsilon_{i2}, \ \varepsilon_{i2} \sim \mathcal{N}(0, 1), \quad i = 5, 6, 7, 8,$$
$$X_i = V_3 + \varepsilon_{i3}, \ \varepsilon_{i3} \sim \mathcal{N}(0, 1), \quad i = 9, 10,$$

$$\varepsilon_{ij} \text{ are independent } \ i = 1, 2, \ldots, 10; \ \ j = 1, 2, 3.$$

We take $\mathbf{A}$ to be the exact covariance matrix of $(X_1\ X_2\ \ldots\ X_{10})$ to compute the top principal component. As the first two factors *i.e.*, $V_1$ and $V_2$ are associated with four variables while the last one *i.e.*, $V_3$ is associated with only two variables and noting that all the three factors $V_1$, $V_2$, and $V_3$ roughly have the same variance, $V_1$ and $V_2$ are almost equally important, and they are both significantly more important than $V_3$. Therefore, for the first sparse principal component, the ideal solution would be to use either $(X_1, X_2, X_3, X_4)$ or $(X_5, X_6, X_7, X_8)$. Using the true covariance matrix and the oracle knowledge that the ideal sparsity is $k = 4$, we apply our algorithms and compare it with spca of (Zou et al., 2006) as well as the SDP-based algorithm of (d'Aspremont et al., 2007). We found that while two of our methods, namely, spca-d (Algorithm 3) and spca-sdp (Algorithm 5) are able to identify the correct sparsity pattern of the optimal solution, spca-r (Algorithm 2) wrongly includes the variable $X_{10}$ instead of $X_8$, possibly due to high correlation between $V_2$ and $V_3$ (see Table 2 for details). However, the output of the spca-d is much more interpretable, even though it has slightly lower PVE than spca-r.

In our additional experiments on the large datasets, Figure 2b shows the performance of various SPCA algorithms on the synthetic data. Notice that the performance of the maxcomp heuristic is worse than spca as well as our algorithms. This is quite evident from the way we constructed the synthetic data. In particular, turning the bottom $\frac{n}{4}$ elements of $\tilde{\mathbf{V}}$ into large values guarantees that these would not be good elements to retain in the construction of the output vector in maxcomp, as they fail to capture the right sparsity pattern. On the other hand, our algorithms perform better than or comparable to spca. Similar to the real data, performance of spca-sdp closely matches with that of dec, cwpca, and spca-lowrank. In Figure 3, we demonstrate how our algorithms perform on CHR 3 and CHR 4 of the population genetics data. We see a similar behavior as observed for CHR 1 and CHR 2 in Figures 1a-1b. In Table 3, we report the variance $f(\mathbf{y})$ captured by the output vectors of different methods for the text data, which again validates the accuracy of our algorithms.

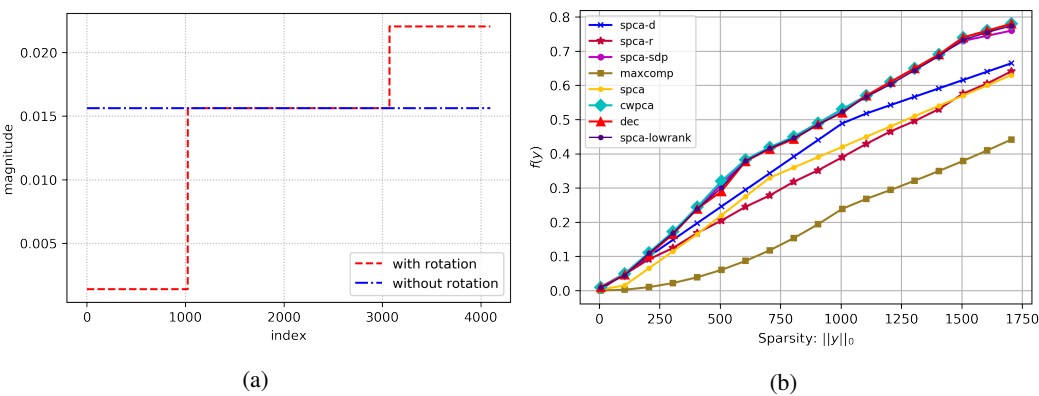

(a)  (b)

Fig. 2: Experimental results on synthetic data with $m = 2^7$ and $n = 2^{12}$: (a) the red and the blue lines are the sorted absolute values of the elements of the first column of matrices $\mathbf{V}$ and $\tilde{\mathbf{V}}$ respectively. (b) $f(\mathbf{y})$ vs. sparsity ratio.

| $k$ | pca | spca | maxcomp | spca-d | spca-r | spca-sdp | cwpca | dec | spca-lowrank |
|---|---|---|---|---|---|---|---|---|---|
| 100 | 0.4597 | 0.0774 | 0.0309 | 0.1385 | 0.0247 | 0.1644 | 0.1519 | 0.1379 | 0.1556 |
| 500 | 0.4597 | 0.1874 | 0.2140 | 0.2948 | 0.1683 | 0.3266 | 0.3542 | 0.3378 | 0.3511 |
| 1,000 | 0.4597 | 0.2737 | 0.3625 | 0.3650 | 0.2892 | 0.4128 | 0.4331 | 0.4214 | 0.4306 |
| 2,000 | 0.4597 | 0.4056 | 0.4380 | 0.4334 | 0.3935 | 0.4396 | 0.4554 | 0.4408 | 0.4545 |
| 5,000 | 0.4597 | 0.4257 | 0.4441 | 0.4422 | 0.4276 | 0.4413 | 0.4558 | 0.4521 | 0.4560 |
| 10,000 | 0.4597 | 0.4462 | 0.4512 | 0.4505 | 0.4412 | 0.4485 | 0.4559 | 0.4543 | 0.4571 |

Table 3: Text data: $f(\mathbf{y})$ vs. the sparsity parameter $k$ for various SPCA algorithms.

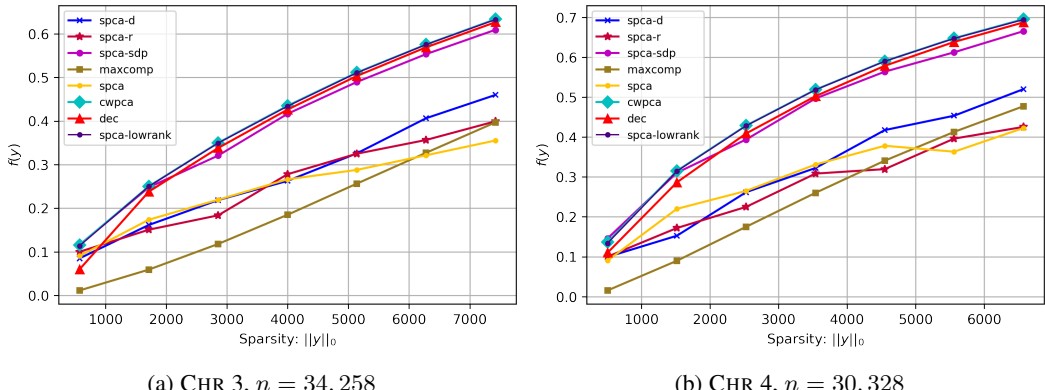

(a) CHR 3, $n = 34,258$           (b) CHR 4, $n = 30,328$

Fig. 3: Experimental results on real data: $f(\mathbf{y})$ vs. sparsity ratio.

Table 4: Statistics of the population genetics data.

| Dataset | # Rows ($m$) | # Columns ($n$) | Density |
|---------|-------------|-----------------|---------|
| CHR 1 | 2,240 | 37,493 | 0.986 |
| CHR 2 | 2,240 | 40,844 | 0.987 |
| CHR 3 | 2,240 | 34,258 | 0.986 |
| CHR 4 | 2,240 | 30,328 | 0.986 |

Table 5: Statistics of gene expression and text data.

| Dataset | # Rows ($m$) | # Columns ($n$) | Density |
|---------|-------------|-----------------|---------|
| Gene expression | 107 | 22,215 | 0.999 |
| Text classification | 2,858 | 12,427 | 0.004 |

Table 6: Values of $\sigma_1(\mathbf{Z})$, $\kappa(\mathbf{Z})$, $\alpha$, and $\gamma_Z$ for various datasets.

| Dataset | $\sigma_1(\mathbf{Z})$ | $\kappa(\mathbf{Z})$ | $\alpha$ | $\gamma_Z$ |
|---------|------------------------|----------------------|----------|------------|
| Pit Props with $k = 7$ | 0.9999 | $3.95 \times 10^{10}$ | 1.0000 | 0.9999 |
| Data of Zou et al. (2006) with $k = 4$ | 0.9999 | $5.14 \times 10^{10}$ | 1.0000 | 0.9999 |
| CHR 1 with $k = 10,000$ | 0.9985 | $7.12 \times 10^{6}$ | 1.0017 | 0.9968 |
| CHR 2 with $k = 10,000$ | 0.9996 | $6.94 \times 10^{6}$ | 1.0003 | 0.9982 |
| CHR 3 with $k = 10,000$ | 0.9995 | $9.47 \times 10^{7}$ | 1.0005 | 0.9989 |
| CHR 4 with $k = 10,000$ | 0.9998 | $1.27 \times 10^{7}$ | 1.0002 | 0.9991 |
| Gene expression with $k = 5,000$ | 0.9913 | $2.05 \times 10^{6}$ | 1.0001 | 0.9914 |
| Text Classification with $k = 5,000$ | 0.9997 | $5.78 \times 10^{6}$ | 1.0001 | 0.9987 |

