# OpenReview forum: "Approximation Algorithms for Sparse Principal Component Analysis"
_ICLR.cc/2021/Conference — Reject_

### Official Review · AnonReviewer1 · 2020-10-26
**good paper**

**Rating:** 7
**Confidence:** 4

**Review:**

This paper proposed three simple algorithms for sparse principal component analysis (SPCA): a) randomized matrix multiplication; b) deterministic thresholding scheme; and c) semidefinite programming relaxation.
All of the proposed algorithms look like native combinations of existing techniques and simple sparsification steps. However, it is somewhat surprising that such simple strategies have reasonable theoretical guarantees whose error bounds depend on the properties of input matrix and the target sparsity.

I have some comments as follows:

1. The analysis of Algorithm 2 and 3 are based on full SVD whose complexity is $O(n^3)$. The authors also mentioned that we can use iterative methods to replace full SVD in practice.  Hence, I think it is necessary to provide the theoretical analysis for the proposed frameworks with inexact SVD and establish the error bounds contain the error from approximation of SVD. Similarly, it is also desired to give the results for Algorithm 5 when we only obtain an approximate solution of problem (2).

2. It is prefer to test the proposed algorithms on large scale datasets, which could make the paper be more convincing to machine learning community. Additionally, since  $O(n^3)$ is unacceptable for large $n$, it is necessary to implement the ideas of this paper with inexact SVD and report the corresponding empirical results.

---

> ### Author Response · Authors · 2020-11-25
> **Response to AnonReviewer1**
>
> We thank the reviewer for their positive assessment of the paper. We agree that analysis for approximate solutions from the SVD and SDP procedures would be comprehensive for our algorithms. We have included the analysis for both approximate SVD and SDP solutions in our revised version.

---

### Official Review · AnonReviewer2 · 2020-10-28
**Need clarity and experimental evidence on the tightness of the bound.**

**Rating:** 4
**Confidence:** 3

**Review:**

This paper proposed three simple algorithms for sparse principal component analysis (SPCA): a) randomized matrix multiplication; b) deterministic thresholding scheme; and c) semidefinite programming relaxation. All of the proposed algorithms look like native combinations of existing techniques and simple sparsification steps. However, it is somewhat interesting to have novel theoretical guarantees for these simple strategies whose error bounds depend on the properties of the input matrix and the target sparsity.

This paper is mathematically sound and the theoretical bound of existing SPCA tricks is also interesting to know. But the significance of the work is not clearly written. Novelty is marginal as these tricks were already there it seems. It needs clarity on improvement on the bound in terms of assumptions and tightness of existing bounds related to SPCA provided by d’Aspremont et al., 2014, Papailiopoulosetal.,2013

The experiment section is weak as the author did not compare to recent methods like d’Aspremont et al., 2014, Papailiopoulosetal.,2013, The result achieved by artificial data is not demonstrating the significance of the proposed method in comparison with the existing methods. (supplementary material)

No experimental validation is given on the bound. It will be good to verify how much tight these theoretical bounds are.

The analysis of Algorithm 2 and Algorithm 3 are based on that of full SVD whose complexity is  O(n^3).  It will be nice to know what is the running time for the proposed algorithm on a moderately large dataset.

I recommended a reject as it has marginal novelty and the claims of the proposed method are not experimentally well supported in the paper.

---

> ### Author Response · Authors · 2020-11-25
> **Response to AnonReviewer2**
>
> We thank the third reviewer for their detailed feedback. We agree that more thorough comparison to the works of  [d’Aspremont et al., 2014] and [Papailiopoulos et al., 2013] would have been helpful in distinguishing our results. Namely, [d’Aspremont et al., 2014] evaluates the SDP relaxation of the problem without theoretical guarantees on the quality of the output, whereas we give provable and explicit guarantees on the quality of our output. Similarly, the results of [Papailiopoulos et al., 2013] are expressed in terms of a ratio of eigenvalues. Thus their theoretical guarantees have limited context without any assumptions on the ratio of eigenvalues in the input. Our results do not require such an assumption.
>
> In the updated version, along with other relevant methods, we also included a comparison of our algorithms with the spannogram-based method of [Papailiopoulos et al., 2013] and found that our Algorithm 5 indeed results in a solution very similar to their output. However, we also note that in larger datasets, to get a highly accurate output from [Papailiopoulos et al., 2013], one typically needs d > 3, which makes the implementation significantly slow.
>
> On the other hand,  in our context, [d’Aspremont et al., 2014] did not seem to provide any explicit implementation that returns a k-sparse vector and therefore, we did not have any algorithmic results for a direct comparison.
>
> Using the pitprops dataset, we plotted the lower bounds of our theoretical guarantees along with [Papailiopoulos et al., 2013] to verify the tightness of our bounds. We assume epsilon=0.1 and found that the lower bound of our SDP-based method (Algorithm 5) is indeed very close to that of [Papailiopoulos et al., 2013] (with d=3). However as mentioned before, the accuracy parameter of [Papailiopoulos et al., 2013] typically relies on the spectrum of A. For a highly accurate output of [Papailiopoulos et al., 2013], epsilon can be much smaller depending on the structure of A, in which case the difference between the lower bounds of our Algorithm 5 and [Papailiopoulos et al., 2013] is even smaller.
>
> We do note that although the current analyses of our Algorithm 2 and Algorithm 3 are based on exact SVD, in our implementations we have used fast SVD algorithms to speed up the running time. As suggested by AnonReviewer1, we have also included the corresponding analyses using approximate SVD in the revised version.

---

### Official Review · AnonReviewer3 · 2020-10-28
**a paper of theoretical research interest only**

**Rating:** 5
**Confidence:** 3

**Review:**

This paper presents three approximation algorithms for the sparse PCA problem. They are (i) randomized matrix multiplication, (ii) deterministic thresholding scheme, (iii) semidefinite programming relaxation. All the three algorithms have provable theoretical guarantees and low-degree polynomial time.

I have the following comments.
1. The authors propose a set of algorithms. It makes me feel that if any of the algorithms really works, the authors do not necessarily propose three algorithms.

2. The algorithm only output a solution with expected k-sparsity. However, its variance could be large. This significantly limits it practical use since we often need a exactly k sparse solution.

3. The experimental comparisons are not sufficient. The authors should compare their methods with state-of-the-art sparse PCA solvers such as coordinate-wise optimization method (The sparse principal component analysis problem: Optimality conditions and algorithms, JOTA 2016) and block decomposition method (A Decomposition Algorithm for the Sparse Generalized Eigenvalue Problem, CVPR 2019). In addition, the dimension of the data set is too small to show the effectiveness of the algorithm. For the pit props data set which only contains 13 dimensions, it is shown that the block decomposition method can find the global optimal solution.

4. This paper is of theoretical research interest only. ICLR  emphasises a lot on experiments and empirical evaluation, STOC/FOCS/SODA could be a better place for the present paper.

---

> ### Author Response · Authors · 2020-11-25
> **Response to AnonReviewer3**
>
> We thank the reviewer for their thoughtful comments on our paper. We respectfully disagree with the notion that none of our algorithms really work; we instead emphasize the advantages of each algorithm.
>
> Our first algorithm is deterministic, which is often necessary for settings in which randomness is expensive and also interesting for theoretical purposes. For example, deterministic algorithms are sometimes needed in the presence of adaptivity - one cannot use the same randomness for randomized algorithms if you update the matrix in a way that depends on the previous answer, e.g., if the input is chosen adversarially.
>
> Although our second algorithm is randomized, it is simple to implement and very fast in practice. Thus our second algorithm is especially useful in settings where extremely high accuracy is not necessary. Our third algorithm is more complicated than the previous algorithms, but it also performs the best in practice. In fact, our experiments show that our third algorithm is competitive with state-of-the-art algorithms, even though we have not fully optimized its performance.
>
> We agree with the observation that the variance of the sparsity could be large even though the expectation is small. However, note that we can upper bound the probability that the sparsity of a single instance is bad, using Markov's inequality. Thus this concern is addressed by running the algorithm a constant number of times in parallel to boost the probability of success. We have added more formal details into our revision version.
>
> We have also significantly expanded our experimental evaluations in the revised version to include empirical comparisons to the suggested state-of-the-art SPCA solvers  (JOTA 2016 and CVPR 2019). We found that our SDP-based method (Algorithm 5) not only recovers the optimal solution in smaller datasets (pitprops and synthetic data), but also performs competitively with
> the aforementioned state of the art algorithms on larger datasets.
>
> We do emphasize that our empirical evaluations not only serve as a proof-of-concept to verify our theoretical findings, but also in practice, given the comparisons with other relevant methods, we show that our algorithms can provide interpretable sparse PCA and in particular, the output of Algorithm 5 can indeed match with the accuracy of previous methods.

---

### Official Review · AnonReviewer4 · 2020-10-29
**The paper provides three approaches for sparse PCA problem, with theoretical guarantees for the sparsity level and the optimality gap of the output solution.**

**Rating:** 4
**Confidence:** 3

**Review:**

Overall, I vote for weak rejecting. The theoretical findings in this paper is presented clearly and looks solid. However, my major concern is the meaningfulness of the problem (in particular, the setting that imposes no assumption on the input matrix) studied in this paper (see cons below). Hopefully the authors can address my concern in the rebuttal period.


##########################################################################Pros:


1. The paper studies the sparse PCA problem. While no assumptions is imposed on the input covariance matrix, the authors give theoretical guarantees on the optimality gap (i.e. the gap between the optimal value and the objective value achieved by the solution) as well as the sparsity level of the solution.

2. The authors provide numerical experiments to illustrate their theoretical findings for the three presented algorithms.


##########################################################################

Cons:


1. The sparse PCA problem studied in prior literatures aims to estimate the leading eigenvectors of a covariance matrix under the high dimensional setting, when the leading eigenvectors are assumed to be sparse. This paper imposes no constraint on the input covariance matrix. This leads to my major concerns as follows:

(i) Is the problem still meaningful when we do not assume that the leading eigenvectors of the input covariance matrix are not sparse? If this is the case, I think it is not reasonable to seek for a sparse estimate of the leading eigenvector.

(ii) In the case when the top eigenvector is sparse, the theoretical findings in this paper does not provide guarantees that the output of the three algorithms are reliable approximation of the top eigenvector. For Algorithm 2 and Algorithm 3, the results in Theorem 2.2 and 3.1 indicates that there is a nonnegligible optimality gap, which can be as large as half the optimal value. For Algorithm 3, the results in Theorem 4.1 also relies on the properties of the SDP solution (through constants $\alpha$ and $\kappa(Z)$), which also might result in a large optimality gap.

I understand that the main purpose of this paper is to study SPCA problem under no assumptions on the covariance matrix, so it might not be necessary to address (ii). In that case, I suggest the authors provide more explanations for the meaning of solving SPCA problem (equation (1)) for general input covariance matrix, as well as the implications of the theoretical guarantees (e.g. the reason why the outputs of the three algorithms are reliable/meaningful estimates, given the fact that they satisfy the theoretical guarantees shown in the theorems) to help address (i).

2. In Theorem 4.1, the exceptional probability is 3/8. It might be better to set the exceptional probability to be $\delta$, and show the dependency of other quantities on $\delta$.

---

> ### Author Response · Authors · 2020-11-25
> **Response to AnonReviewer4**
>
> We thank the reviewer for their thorough assessment of our paper. Although we make no assumptions on the leading eigenvectors of the covariance matrix, the leading eigenvector is indeed usually sparse in many applications. For example, [d'Orsi, et. al., 2020] assumes that the covariance matrix is formed by an outer product of a single sparse vector and some Gaussian noise. It should be noted that even under these assumptions, [d'Orsi, et. al., 2020] show lower bounds for non-negligible optimality gaps.
>
> Although their gap is significantly better than ours, their results may indicate stronger lower bounds for non-negligible optimality gaps in our setting, where we do not assume any input distribution. Thus an interpretation of our error guarantee is that our algorithm performs well especially when the top eigenvector is sparse, but its performance decays gracefully as the top eigenvector contains more nonzero entries. Moreover, we empirically show that the parameter alpha is Theorem 4.1 is quite small in practice.
>
> As noted after Theorem 2.2, we agree that our analysis obtaining 3/8 probability of success can be generalized to arbitrary 1-delta probability of success. In our revision version, we have added a similar statement after Theorem 4.1 for completeness.
>
> [d'Orsi, et. al., 2020] Sparse PCA: Algorithms, Adversarial Perturbations and Certificates. Tommaso d’Orsi, Pravesh K. Kothari, Gleb Novikov, and David Steurer. FOCS 2020. https://arxiv.org/pdf/2011.06585.pdf

---

### Decision · Program_Chairs · 2021-01-07
**Final Decision**

**Decision:**

Reject

**Comment:**

The paper proposes three algorithms for the sparse PCA problem, where one imposes the additional constraint that the vectors have a small number of non-zero entries. The proposed algorithms run in polynomial time and achieve provable approximation guarantees on the accuracy and sparsity. The reviewers identified the following strengths of the contributions: the algorithms are simple and have different strengths; the theoretical results are sound and perhaps even surprising; the presentation is clear. The reviewers identified the following weaknesses of the contributions: the running times of the proposed algorithms are high and they may not scale to large datasets, which significantly limits their application to machine learning datasets; the experimental evaluation is insufficient and it does not compare with some of the state of the art algorithms; the algorithmic novelty is limited. After weighing these strengths and weaknesses as well as evaluating the paper relative to other ICLR submissions, I recommend reject.

---

> ### Author Response · Authors · 2021-02-23
> **Incorporation of Reviewer Feedback**
>
> We thank the ICLR 2021 Conference Program Chairs for a detailed summary of the reviewers' remarks. For completeness, we offer the following response summarizing our incorporation of the reviewer feedback to improve future versions of our manuscript.
>
> Indeed our algorithms are both simple to understand and simple to implement, with various strengths such as determinism, performance, or runtime. In fact, we have subsequently improved the runtime from polynomial time to near-linear (or near input-sparsity) runtime. This was achieved by incorporating a reviewer's remark that the optimal solution to our SDP is not necessary to achieve our approximation guarantee. Similarly, we only need to compute an approximate SVD rather than a full SVD to achieve our guarantees, which can be done in near input-sparsity runtime rather than matrix multiplication runtime. In summary, our algorithms can be further optimized to use near-linear runtime rather than polynomial runtime.
>
> We have also significantly expanded our experimental evaluations to incorporate empirical comparisons to the suggested state-of-the-art SPCA solvers (JOTA 2016 and CVPR 2019). Our SDP-based approach not only recovers the optimal solution in smaller datasets, but also performs competitively with the aforementioned state of the art algorithms on larger datasets. Given the comparisons with other relevant methods, we show that our algorithms can provide interpretable sparse PCA and in particular, the output of SDP-based algorithm can indeed match with the accuracy of previous methods.